# Sensitive quantification of carbon monoxide in vivo reveals a protective role of circulating hemoglobin in CO intoxication

Qiyue Mao[1], Akira T. Kawaguchi[2], Shun Mizobata[1], Roberto Motterlini [3✉], Roberta Foresti [3✉] & Hiroaki Kitagishi [1✉]

Carbon monoxide (CO) is a gaseous molecule known as the silent killer. It is widely believed that an increase in blood carboxyhemoglobin (CO-Hb) is the best biomarker to define CO intoxication, while the fact that CO accumulation in tissues is the most likely direct cause of mortality is less investigated. There is no reliable method other than gas chromatography to accurately determine CO content in tissues. Here we report the properties and usage of hemoCD1, a synthetic supramolecular compound composed of an iron(II)porphyrin and a cyclodextrin dimer, as an accessible reagent for a simple colorimetric assay to quantify CO in biological samples. The assay was validated in various organ tissues collected from rats under normal conditions and after exposure to CO. The kinetic profile of CO in blood and tissues after CO treatment suggested that CO accumulation in tissues is prevented by circulating Hb, revealing a protective role of Hb in CO intoxication. Furthermore, hemoCD1 was used in vivo as a CO removal agent, showing that it acts as an effective adjuvant to $O_2$ ventilation to eliminate residual CO accumulated in organs, including the brain. These findings open new therapeutic perspectives to counteract the toxicity associated with CO poisoning.

[1] Department of Molecular Chemistry and Biochemistry, Faculty of Science and Engineering, Doshisha University, Kyotanabe, Kyoto, Japan. [2] Cell Transplantation and Regenerative Medicine, Tokai University, Isehara, Kanagawa, Japan. [3] University Paris Est Creteil, INSERM, IMRB, Creteil, France. ✉email: roberto.motterlini@inserm.fr; roberta.foresti@inserm.fr; hkitagis@mail.doshisha.ac.jp

Carbon monoxide (CO) is a diatomic gaseous molecule produced by incomplete combustion of carbon-based materials and is mostly known for its toxic properties when inhaled in high amounts. CO intoxication, which frequently occurs in suicides, during fires, and other accidents, is the most common type of chemical poisoning with an increased risk of death[1]. In spite of its notorious toxicity, CO is relatively unreactive compared to molecular oxygen ($O_2$), nitric oxide (NO), hydrogen sulfide ($H_2S$), and reactive oxygen species (ROS) such as superoxide ($O_2^{\bullet-}$), hydroxyl radical ($\bullet OH$), hydrogen peroxide ($H_2O_2$), and singlet oxygen ($^1O_2$). Although other small molecules show various reactivities toward proteins, lipids, and nucleic acids, the only recognized reaction of CO in the biological system is the binding to low valent metal ions, mostly ferrous heme[2–5]. Due to a specific interorbital interaction, called π-backbonding or π-backdonation, the carbon atom of CO forms a particularly strong bond with low valent metal ions such as iron (II)heme with a high stability and high binding affinity[6,7]. Formation of the stable heme-CO complex in vivo is the cause of CO toxicity; once CO is inhaled, it strongly binds to heme in hemoproteins that are indispensable for carrying or consuming $O_2$, such as hemoglobin (Hb), myoglobin (Mb), and cytochrome $c$ oxidase (CcO) among others, leading to their dysfunction[8,9]. Levels of CO-bound Hb (CO-Hb) in blood are considered an important biomarker to assess CO poisoning[1,10,11]. CO-Hb below 10% normally circulates in blood due to endogenously produced CO or CO from cigarette smoking, with no apparent clinical symptoms. When CO-Hb reaches levels between 20 and 40% due to CO gas inhalation, symptoms such as dizziness, headache, and nausea begin to appear. CO-Hb values exceeding 40% increase the risk of death. Since the binding of CO to heme is thermodynamically controlled and thus reversible, CO-Hb levels can be reduced by excess $O_2$. Normobaric[12,13] and hyperbaric $O_2$ ventilations[14,15] are thus current major approaches for treating CO intoxication.

A unique study conducted in dogs and reported in 1975[16] provides an important insight on the mechanism of CO poisoning. When dogs received CO gas by inhalation, they all died rapidly within 1 h with CO-Hb levels of 54–90%. In contrast, if the animals were first bled to reduce blood volume and then transfused with red blood cells (RBC) saturated with CO, all dogs survived despite reaching CO-Hb levels of 60%[16]. These findings strongly indicate that inhalation of gaseous CO is highly toxic whereas CO-Hb is not, ultimately challenging the commonly accepted notion that the percentage of CO-Hb is the appropriate marker to establish the toxicity of CO inhalation. Indeed, patients can die of CO poisoning even with CO-Hb saturation levels below 30%[15,17], indicating that crucial toxicity derives from gaseous CO diffusing into organs/tissues, directly compromising the activities of key hemoproteins such as CcO that requires $O_2$ to produce energy[8]. From the viewpoints of medical diagnosis and toxicology, knowing how much CO is accumulated in organs/tissues in subjects affected by CO intoxication would be fundamentally important.

Detection and quantification of CO contained in biological samples have been a long-standing challenge. In medical practice, CO-Hb in blood is measured by oximeters, which rely on a colorimetric assay to diagnose CO intoxication[10]. In addition, gas chromatography (GC) techniques are the main methodology used to detect CO contained in tissues, as reported by the laboratories of Coburn and Vreman to determine endogenous CO distributed in mice[18] and human tissues[19,20]. GC is considered to be well established as a CO quantifying method[10,21]. However, as we investigated and discuss in this paper, the headspace analysis by GC tends to underestimate the amount of CO, highlighting that a convenient and reliable method for measuring CO contained in organs/tissues is needed.

Biomimetic chemistry for hemoproteins started in the 1970s. An epoch-making model for $O_2$-binding hemoproteins is a picket-fence porphyrin synthesized by Collman et al.[22–24]. After their discovery, many synthetic analogs were synthesized to demonstrate the reversible $O_2$/CO bindings in anhydrous organic solvents[23,24]. The weakness of these synthetic model systems was that even a trace amount of water contaminating the solvent was unacceptable due to a water-catalyzed autoxidation of the iron(II) to iron(III)porphyrins[25,26]. Because of the difficulty in preparing a hydrophobic heme pocket similar to native hemoproteins, aqueous biomimetic compounds have been scarce[27,28]. Nevertheless, our laboratory has succeeded in preparing hemoprotein models working in 100% aqueous solutions[29–35]. In our model system, the iron(II)porphyrin is embedded in hydrophobic per-O-methyl-β-cyclodextrin (CD) cavities, thus avoiding a water-catalyzed autoxidation even in 100% water at ambient temperatures. As a result, stable and reversible $O_2$ and CO bindings similar to native Hb and Mb have been achieved.

Among the hemoprotein model complexes synthesized in our laboratory, hemoCD1 (Fig. 1A) composed of 5,10,15,20-tetrakis (4-sulfonatophenyl)porphinatoiron(II) ($Fe^{II}TPPS$) and a per-O-methyl-β-cyclodextrin dimer with a pyridine linker (Py3CD), showed the highest CO binding affinity ($P_{1/2}^{CO} = 1.5 \times 10^{-5}$ Torr, $K_d = 19.2$ pM in phosphate buffer at pH 7 and 25 °C)[30]. This is ~100 times higher than that of Hb in R-state (Hb-R), whereas the $O_2$ binding affinity of hemoCD1 is moderate ($P_{1/2}^{O2} = 10$ Torr, $K_d = 17$ μM in phosphate buffer at pH 7 and 25 °C)[30,36], close to that of Hb in the T-state (Hb-T). To the best of our knowledge, the CO binding affinity of hemoCD1 is the highest among the reported native CO-binding proteins. The administration of hemoCD1 to animals (rats and mice) did not show any significant side effect except for endogenous CO scavenging[37–40]. Most of the administered hemoCD1 is excreted in the urine as CO-hemoCD1 complex within 1 h. Therefore, we have focused on studying hemoCD1 as a CO scavenging agent in vivo and recently described the effects of endogenous CO-depletion on heme oxygenase expression[38], ROS production[41], and circadian clock gene expressions[40]. For years we have recognized that hemoCD1 not only is a powerful tool for elucidating the biological role of endogenous CO but could also be useful for detecting and removing excessive CO in the living organisms.

In this paper, we established a new method for CO detection and quantification using hemoCD1 in rat tissues. The binding of CO to hemoCD1 is determined by a simple colorimetric assay using absorbances at three different wavelengths (422, 427, and 434 nm) and was evaluated in rats exposed to CO gas compared to untreated animals. Importantly, kinetic studies on biodistribution of CO in tissues after CO gas inhalation suggest a protective effect of circulating Hb in CO intoxication. Finally, we describe the potential use of hemoCD1 as an antidote for CO poisoning in mammals.

## Results

**HemoCD1 as a suitable and effective CO scavenger**. The structure of hemoCD1 is shown in Fig. 1. In the presence of $Na_2S_2O_4$, the oxidized form of hemoCD1 (met-hemoCD1) is converted to deoxy-hemoCD1 (Fig. 1A), with a typical absorption band at 434 nm (Fig. 1B). Upon addition of CO gas, deoxy-hemoCD1 is smoothly converted into its CO-bound form, CO-hemoCD1 (Fig. 1A), which has a distinct absorption band at 422 nm (Fig. 1B).

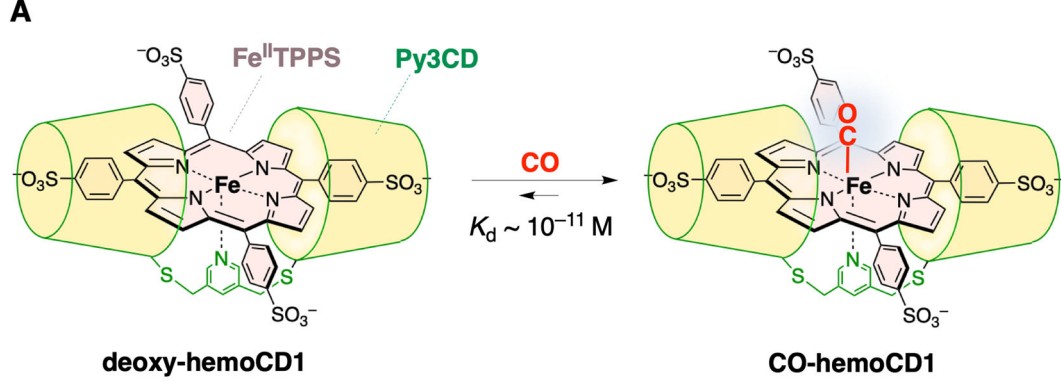

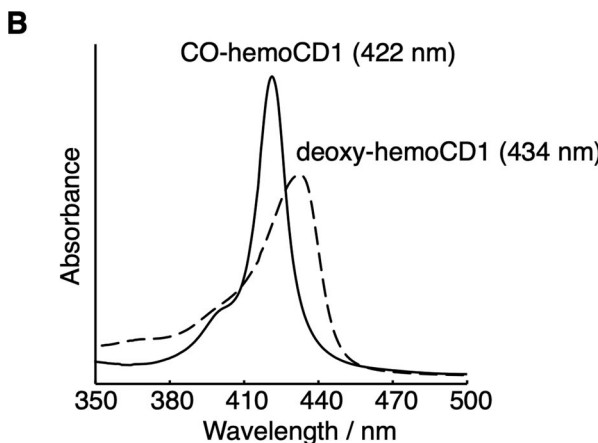

**Fig. 1 HemoCD1, a CO detecting agent. A** HemoCD1 is composed of 5,10,15,20-tetrakis(4-sulfonatophenyl)porphinatoiron(II) (Fe$^{II}$TPPS) and a per-O-methyl-β-cyclodextrin dimer having a pyridine linker (Py3CD). The structure of deoxy-hemoCD1 and CO-hemoCD1 complexes is shown. **B** UV–vis spectra of hemoCD1 showing the Soret bands typical of deoxy-hemoCD1 (434 nm) and CO-hemoCD1 (422 nm) in PBS at pH 7.4 and 25 °C.

Once formed, CO-hemoCD1 is quite stable. In fact, the absorption band of CO-hemoCD1 was unchanged when the solution was bubbled with pure O$_2$ for 5 min (Fig. 2A). On the other hand, CO-Hb in solution was easily converted to its O$_2$-bound form following addition of O$_2$ (Fig. 2B). Once formed, CO-hemoCD1 could not be reconverted to its deoxy form despite continuous N$_2$ bubbling for 60 min, while CO-Hb returned to its deoxy form after N$_2$ treatment (Supplementary Fig. S1). In addition, CO-hemoCD1 remained stable even after the solvent was almost evaporated under high vacuum (<10 Torr) over 2 h. The re-dissolved complex still showed the distinct and typical spectrum of CO-hemoCD1 (Fig. 2C). In contrast, CO-Hb significantly decomposed after the same treatment (Fig. 2D). CO-hemoCD1 was also resistant against excess H$_2$O$_2$, whereas CO-Hb was degraded by H$_2$O$_2$ under the same conditions (Supplementary Fig. S2). CO-hemoCD1 is also stable over 24 h in the presence of biocomponents from cell lysates (Supplementary Fig. S3). These data demonstrate the high stability of the CO-hemoCD1 complex and emphasize that, once bound to hemoCD1, CO hardly dissociates even under these extreme conditions.

To demonstrate the CO-scavenging ability of hemoCD1, CO transfer from CO-Hb to oxy-hemoCD1 (an O$_2$-adduct of hemoCD1) was investigated. The spectra of each component are shown in Fig. 2E and we note the peak in absorbance at 575 nm characteristic of oxy-Hb. After mixing the solutions of oxy-hemoCD1 (0.75 mM, 20 μl in air-saturated buffer) and CO-Hb (0.72 mM, 10 μl in CO-saturated buffer where [CO] = 0.96 mM[42]) in 3 ml air-saturated buffer, a time-dependent increase in the absorbance at 575 nm was observed due to the conversion of CO-Hb to oxy-Hb (Fig. 2F). The two controls, namely (1) the CO-Hb solution without (w/o) oxy-hemoCD1, and (2) the mixed solution of oxy-Hb with CO-hemoCD1, did not reveal any change in absorbance, as shown in Fig. 2F. The first-order CO transfer rate constant was determined (Fig. 2G), which is consistent with the dissociation rate constant ($k_{off}^{CO} = 0.01$ s$^{-1}$) reported for CO-Hb[43]. In the CO transfer reaction, dissociation of CO from CO-Hb is the rate-limiting step, as the on rate for CO to hemoCD1 is much faster ($k_{on}^{CO} = 4.7 \times 10^7$ M$^{-1}$ s$^{-1}$)[30]. After ultrafiltration of the CO-Hb/oxy-hemoCD1 mixed solution using a filter of molecular weight cut off = 30,000 Da, the hemoCD1 and Hb components were separated. The filtrate contained CO-hemoCD1 whereas the residue contained oxy-Hb (Supplementary Fig. S4).

Concerning NO, we have already shown that hemoCD1 is unable to bind NO in the presence of Hb, due to its lower NO binding affinity compared to Hb[41,43]. In addition, CO-hemoCD1 was not decomposed by the addition of NO, whereas the CO in CO-Hb was replaced by NO leading to formation of met-Hb (Supplementary Fig. S5). The stability of CO-hemoCD1 even in the presence of NO is an additional advantage for a selective and accurate detection of CO in vivo. As for H$_2$S, we previously reported that the coordination strength of H$_2$S to iron(II) in hemoCD1 is relatively weak, and the SH$^-$ ligand is easily replaceable with CO in the hemoCD1 analog[44].

The intrinsic toxicity of the CD cavity, which was suggested in a recent article[8], was not observed because the CD cavity is occupied by Fe$^{II}$TPPS in hemoCD1 (Supplementary Fig. S6). We have already reported that the hemoCD1 components have no

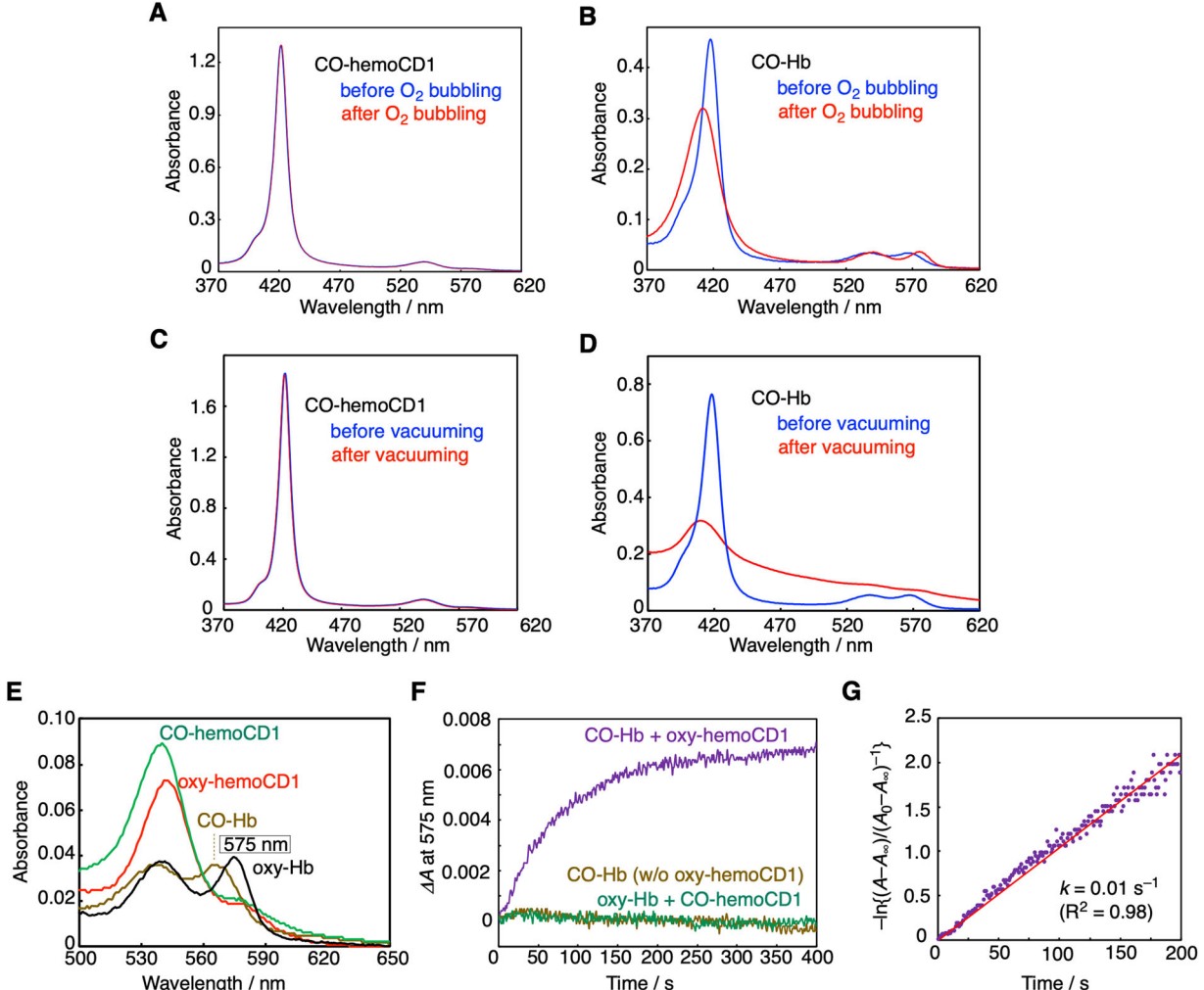

**Fig. 2 Characteristics of hemoCD1 as a CO scavenger.** UV–vis spectra of CO-hemoCD1 (3.5 µM, **A**) and CO-Hb (2.3 µM, **B**) in PBS at pH 7.4 and 25 °C before (blue) and after (red) bubbling O₂ into the solutions for 5 min. CO-Hb was converted to its O₂-bound form (oxy-Hb, red line) whereas CO-hemoCD1 was not. UV–vis spectra of CO-hemoCD1 (5.0 µM, **C**) and CO-Hb (3.7 µM, **D**) before (blue) and after (red) vacuuming under 10 Torr for 2 h followed by re-solubilization of the residues. CO-hemoCD1 was stable while CO-Hb decomposed after this treatment. **E–G** Competition between hemoCD1 and Hb for CO binding. **E** UV–vis spectra of oxy-, and CO-hemoCD1 (5.0 µM each), and oxy- (575 nm absorbance peak) and CO-Hb (2.4 µM each). **F** Time-course for changes in absorbance at 575 nm, indicative of formation of oxy-Hb, after mixing stock solutions of oxy-hemoCD1 (0.75 mM, 20 µl in air-saturated PBS) and CO-Hb (0.72 mM, 10 µl in CO-saturated PBS) in air-saturated PBS (3 ml) at pH 7.4 and 25 °C. Controls are represented by solutions of CO-Hb without (w/o) oxy-hemoCD1, and oxy-Hb mixed with CO-hemoCD1. **G** First-order rate plot for changes in absorbance at 575 nm over time. The linear regression analysis gave a rate constant of $k = 0.01 \, \text{s}^{-1}$.

effect on hemodynamic parameters such as heart rate, blood pressure, and plasma components[37,45]. It has been demonstrated that biological systems do not recognize *meso*-tetraarylporphinatoiron such as FeTPPS in hemoCD1 as a heme cofactor[46,47] and thus hemoCD1 cannot be metabolized by heme oxygenase. Moreover, we demonstrated that injection of oxy-hemoCD1 in rats results in excretion of CO in the urine[37] and that the ferric iron(III) form of the hemoCD derivatives functions as a cyanide-antidote[34,48,49]. Altogether, our data support the idea that hemoCD1 is an effective CO scavenger.

**Development of a new and sensitive CO quantification method using hemoCD1.** Absorbances of the Soret bands of deoxy-hemoCD1 (434 nm) and CO-hemoCD1 (422 nm) were used for CO quantification (Fig. 1B). When aqueous CO solutions with different CO contents were mixed with deoxy-hemoCD1, the Soret band clearly changed as a function of CO showing an isosbestic point at 427 nm (Fig. 3A). Based on the absorbance

ratio at 422 and 434 nm ($A_{422}/A_{434}$), the molar ratio of CO-hemoCD1 in the total hemoCD1 derivatives in the solution ($R_{CO}$) was calculated by the following equation:

$$R^{CO} = \frac{[CO-hemoCD1]}{[deoxy-hemoCD1] + [CO-hemoCD1]}$$
$$= \frac{\varepsilon_{deoxy}^{422} - A_{422}/A_{434} \cdot \varepsilon_{deoxy}^{434}}{A_{422}/A_{434}(\varepsilon_{CO}^{434} - \varepsilon_{deoxy}^{434}) - \varepsilon_{CO}^{422} + \varepsilon_{deoxy}^{422}} \quad (1)$$

where $\varepsilon_{deoxy}^{422}$, $\varepsilon_{deoxy}^{434}$, $\varepsilon_{co}^{422}$, and $\varepsilon_{co}^{434}$ are the molar absorption coefficients of deoxy- and CO-hemoCD1 at 422 and 434 nm, respectively (these coefficients are listed in Table 1). The derivation of Eq. (1) is detailed in Supplementary Information. The total amount of hemoCD1 ($C_{total}$, that is deoxy- and CO-hemoCD1 combined) was determined by the absorbance at 427 nm, the isosbestic point of deoxy- and CO-hemoCD (Fig. 3A), and its absorption coefficient ($\varepsilon^{427}$, Table 1). The amount of CO ($M_{CO}$) in the solution was calculated by multiplying $R_{CO}$ by $C_{total}$.

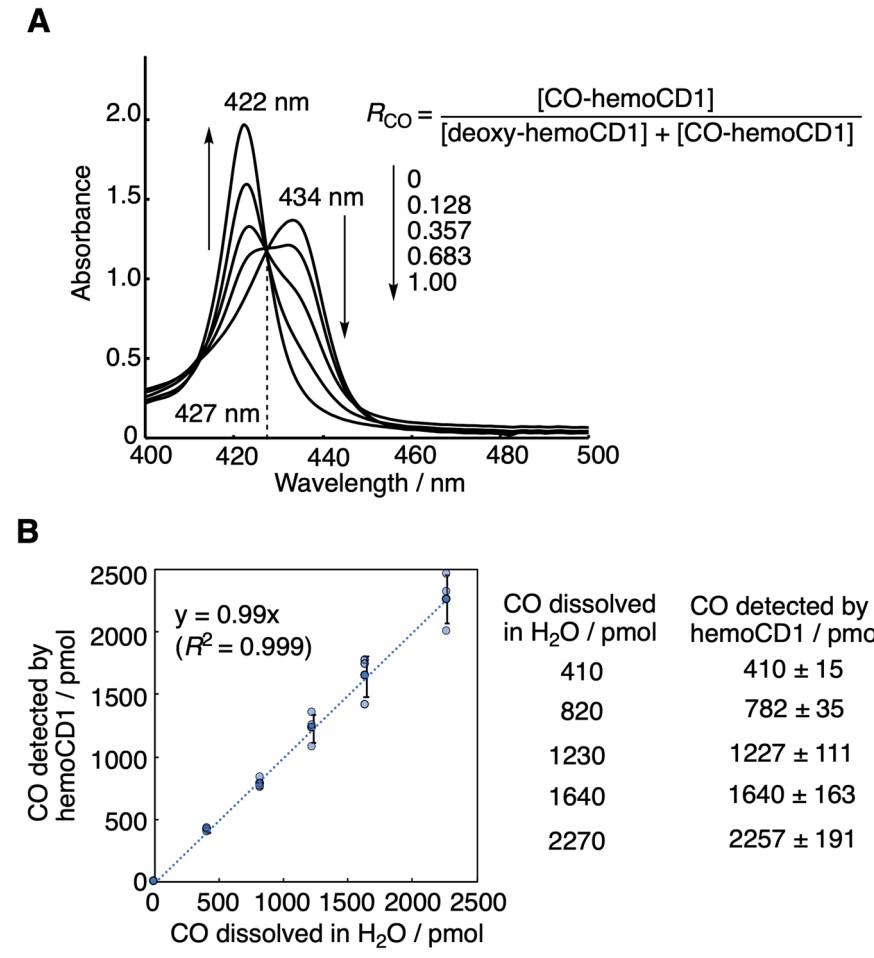

**Fig. 3 Spectroscopic quantification of CO in aqueous solution using hemoCD1. A** UV–vis spectra of hemoCD1 (6.0 µM) after exposure to various amounts of CO in PBS (1 ml) at pH 7.4 and 25 °C. $R_{CO}$ values were calculated using Eq. (1). **B** The plot of known amounts of CO dissolved in water versus the quantified values of CO determined by the hemoCD1 assay. Data are shown as mean ± SD ($n = 3$).

**Table 1 Molar absorption coefficients ($\varepsilon$/M$^{-1}$ cm$^{-1}$) of hemoCD1 used in this study.**

|  | 422 nm | 427 nm | 434 nm |
|---|---|---|---|
| Deoxy-hemoCD1 | 152,000[a] | 195,000[a] | 213,000[b] |
| CO-hemoCD1 | 371,000[b] | 195,000[a] | 67,500[a] |

[a]This work.
[b] 33,41.

We validated our approach by determining the $M_{CO}$ values of known CO standards prepared by adding pure CO gas from a gas-tight syringe to water contained in a rubber-capped bottle without headspace. As shown in Fig. 3B, the amount of CO quantified using hemoCD1 is almost identical to the CO content in the standards ($y = 0.99x$, $R^2 = 0.999$). Due to the limit of absorption detection, ~400 pmol was the minimum value for detection of CO. The slope of the standard curve was 1 with intercept 0; therefore, the amount of CO in solution can be directly determined using Eq. (1) without preparing a standard curve for each detection assay.

**Using the new hemoCD1 assay to quantify endogenous CO levels in tissues**. We next applied our new assay to determine the amount of endogenous CO contained in rat tissues. Samples (5–20 mg) collected from different organs/tissues were homogenized in PBS (0.5 ml). Then, deoxy-hemoCD1 was added to the

tissue suspension followed by sonication (10 s × 2) (see Fig. 4A for methodological details). After sonication, samples were centrifuged for 15 min to obtain a clear supernatant (see picture in Fig. 4A), which was filtered through a 0.45 µm filter. Na$_2$S$_2$O$_4$ added in excess caused denaturation and precipitation of biocomponents that could be readily removed by centrifugation and filtration. Samples were then measured by UV–vis absorption spectrometry, yielding a clear Soret band as exemplified by the liver sample shown in Fig. 4B (red line). Controls were treated in the same manner without adding deoxy-hemoCD1 and their absorbances at 422 and 434 nm (Fig. 4B, blue line) were subtracted from the absorbances of samples added with deoxy-hemoCD1. From these values, $R_{CO}$ was calculated using Eq. (1). The total content of hemoCD1 ($C_{total}$) determined from the absorbance of the isosbestic point at 427 nm was consistent with the amount of deoxy-hemoCD1 initially added to the sample. This indicates that hemoCD1 was not lost during sample treatment, i.e., during sonication, centrifugation, and filtration. From $R_{CO}$ and $C_{total}$, the amount of CO contained in the tissues ($M_{CO}$) was determined.

Because tissue samples contain blood, some residual CO-Hb could still be present in our samples leading to overestimation of CO levels in tissues. Therefore, we compared the CO content without and with flushing organs and tissues with saline. As shown in Figs. 4C and S7, along with a discoloring of tissues as flushing was conducted, the amount of detected CO decreased (see Fig. 4C for liver, and Supplementary Fig. S7 for lung, muscle,

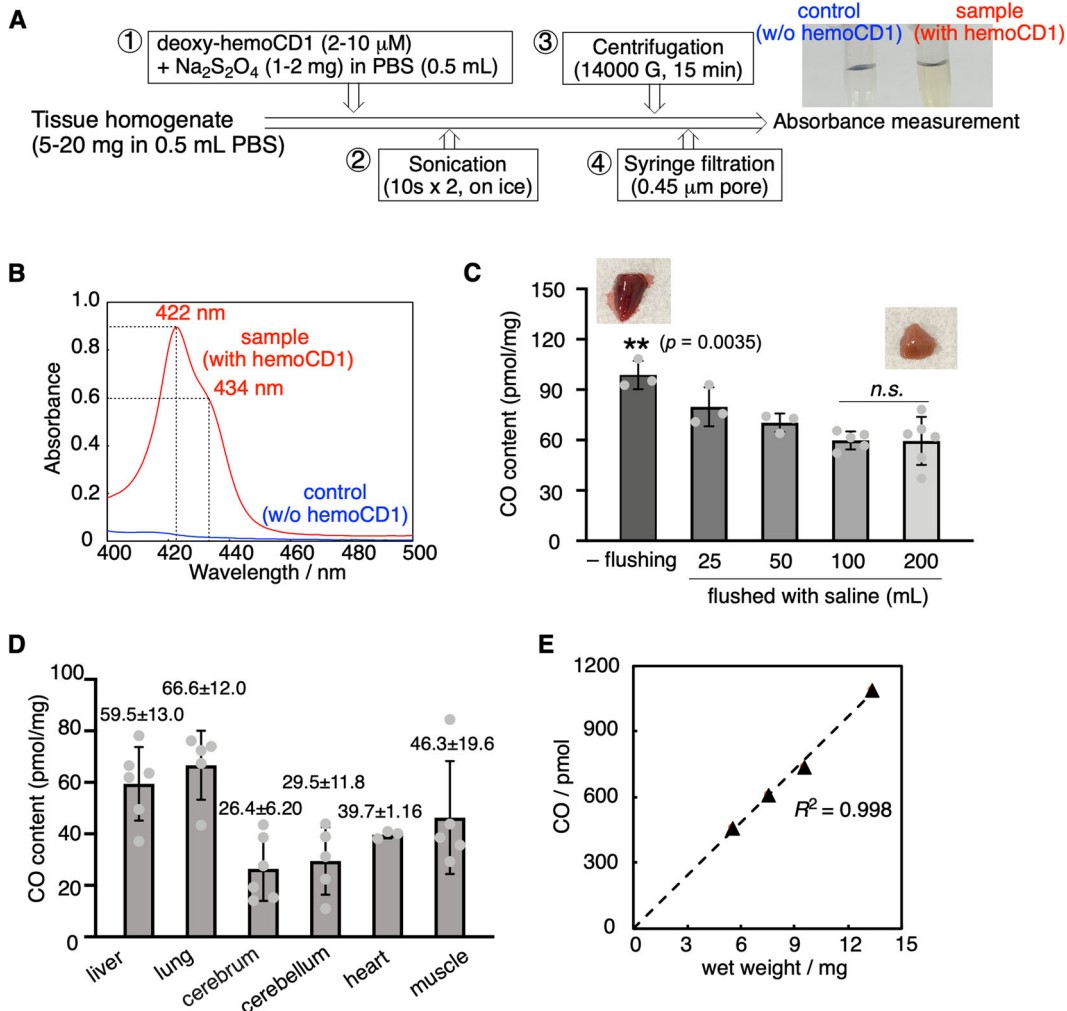

**Fig. 4 Quantification of CO in rat tissues. A** Experimental procedure describing the various steps of the hemoCD1 assay for measuring CO in tissue samples collected from different rat organs. The picture shows clear supernatant solutions of tissue sonicates without (control) and with deoxy-hemoCD1. **B** Typical representative spectra of supernatant solutions of liver sample and control obtained at the end of the hemoCD1 assay. **C** Amounts of CO quantified in liver tissues without (–) or following flushing with 25–200 ml saline. Each bar represents the mean ± SD ($n = 3$–6). Statistical significance, **$p < 0.01$ versus 200 ml flushed organs; n.s. not significant. **D** Content of endogenous CO (pmol/mg, wet weight) detected in different organs using the hemoCD1 assay. Each bar represents the mean ± SD ($n = 6$ for liver, $n = 5$ for lung, $n = 6$ for cerebrum, $n = 5$ for cerebellum, $n = 3$ for heart, and $n = 5$ for muscle). **E** Plot of the wet weight of liver tissue versus the amount of CO detected.

and brain). Even though the CO content was unchanged after flushing with 100 or 200 ml saline, 200 ml saline was chosen to ensure complete flushing of organs. The varying amounts of endogenous CO (pmol/mg tissue, wet weight (ww)) measured in the different organs are reported in Fig. 4D. The amount of CO showed a linear correlation with the mass of tissues (Fig. 4E and Supplementary Fig. S8), supporting the accuracy of the method quantifying CO. As for the spleen, the amount of CO could not be accurately estimated because of interference with endogenous heme-related pigments which could not be adequately removed by the treatment with $Na_2S_2O_4$ and affected the hemoCD1 spectrum. Because hemoCD1 is stable at different pH (pH 4–10)[30], under room light, and in the presence of biological reactive species such as ROS, NO, $H_2S$, and glutathione, the amount of CO in tissues quantified by hemoCD1 was unaffected by these external factors (Fig. S9).

We also compared our CO detection assay with a GC technique equipped with a thermal conductivity detector (TCD), following a methodology described in the literature[18]. In GC analysis, processed tissue samples need to be oxidized by sulfosalicylic acid to liberate CO before measuring gaseous CO in

the headspace[18–21]. As shown in Supplementary Fig. S10, 17.5 ± 0.6 and 12.1 ± 3.1 pmol/mg of CO were detected in liver tissue without or with flushing of the organ with 200 ml saline, respectively. The amount of CO detected in tissue by GC was consistent with data previously reported[50]. On the other hand, we detected considerably more CO using hemoCD1 (98.7 ± 17.5 and 57.8 ± 3.0 pmol/mg). The underestimation of CO by the GC method is probably due to residual CO that is strongly bound to the tissue and can be hardly released by the denaturating agent. The assay using hemoCD1 is capable of detecting more CO in tissues than GC, even though there is still no evidence that all of CO in the tissue was detected by the assay.

**Using the hemoCD1 assay to assess the temporal distribution of CO in blood and tissues after administration of CO gas.** CO present in blood and organs was measured at 5, 10, and 20 min after exposure of rats to air containing 400 ppm CO gas by inhalation (see protocol in Fig. 5A and Supplementary Fig. S11). HemoCD1 was used to assess CO in tissues. Blood CO-Hb levels were measured by a blood analyzer[51] using the venous blood

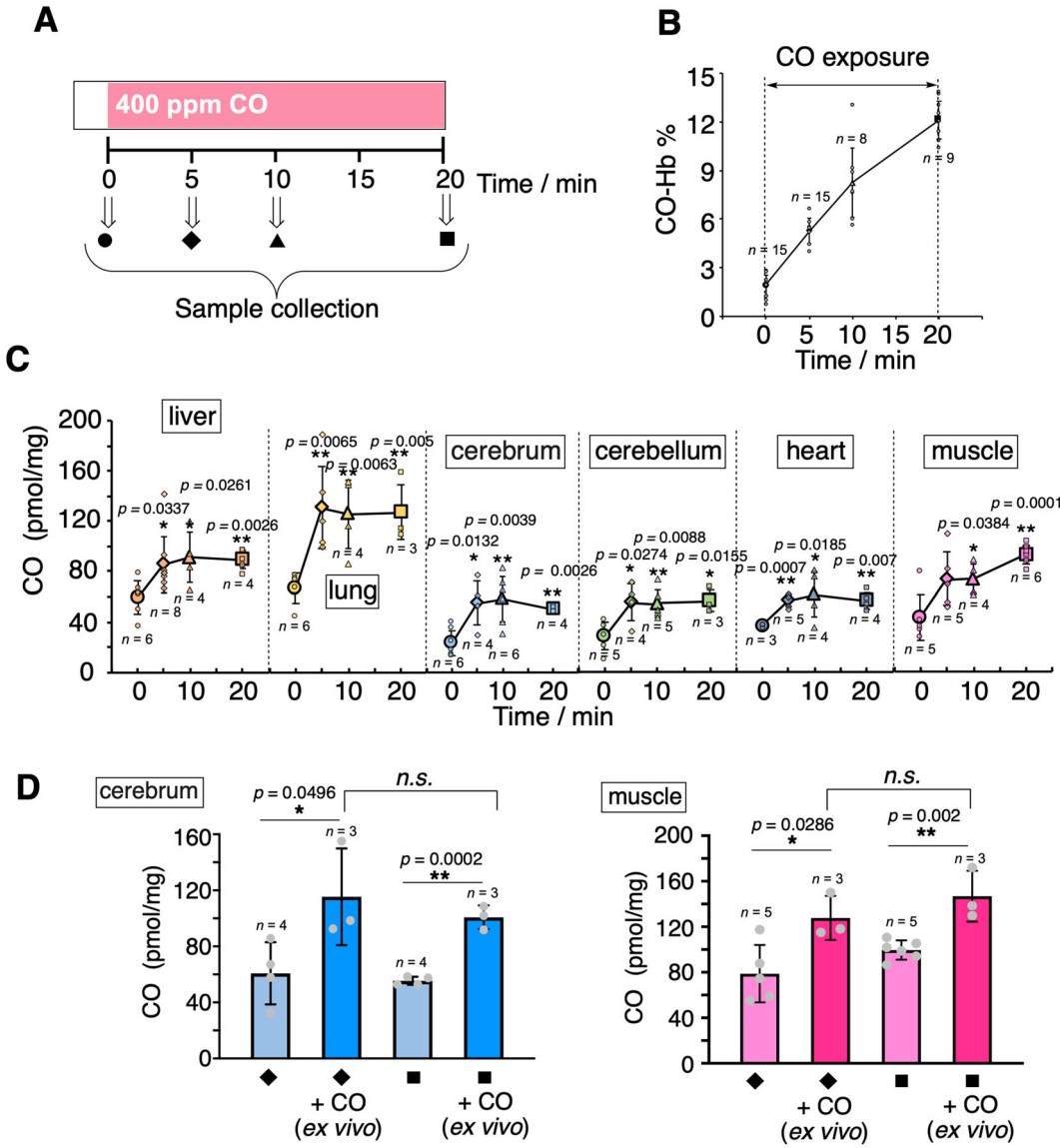

**Fig. 5 Kinetic studies of CO levels in blood and tissues after exposure to CO in rats. A** Anesthetized rats were exposed to CO inhalation (400 ppm) and samples collected at different times as shown. **B** Changes in CO-Hb (%) in the venous blood collected from right ventricles as a function of time. **C** Tissue CO contents as a function of time. Each plot represents the mean ± SD ($n = 3$–6, the number of experiments is shown in the panel). Statistical significance, $*p < 0.05$, $**p < 0.01$ versus $t = 0$. **D** Amount of CO in muscle and cerebrum before and after purging CO gas ex vivo. Samples collected at 5 or 20 min were placed under CO atmosphere for 1 h and then assayed for CO content. Each bar represents the mean ± SD ($n = 3$–5, the number of experiments is shown in the panel). Statistical significance, $*p < 0.05$, $**p < 0.01$; n.s. not significant.

samples collected from right ventricles. As expected, during the 20 min exposure to CO gas, CO-Hb% linearly increased (Fig. 5B). Conversely, the amounts of CO in tissues rapidly increased after 5 min exposure to CO and then reached a plateau at 10 min (Fig. 5C). Thus, it appears that, compared to Hb, the tissue has a limited capacity to store CO. We then took the same tissue samples from rats inhaled with CO gas, placed them ex vivo under a CO atmosphere for 1 h, and quantified the CO content. As shown in Fig. 5D, the tissues could store more CO by adding CO ex vivo. These results indicate that the capacity of the tissues to store CO did not reach saturation during continuous CO inhalation in vivo. Intriguingly, these data also suggest that the high capacity of Hb for CO scavenging during inhalation may confer protection of tissues against CO toxicity. This may be due, among other possibilities, to the ability of Hb to extract CO from the tissues.

To demonstrate this hypothesis, we measured CO accumulated in hepatocytes incubated for 2 h with a CO-releasing molecule

(CORM-401E), followed by replacement of medium in the presence or absence of Hb (Supplementary Fig. S12). Notably, hepatocytes exposed to CORM-401E exhibited a significant increase in intracellular CO, whereas CO levels were equivalent to control values after treatment with CORM401-E followed by oxy-Hb. Spectroscopic analysis revealed that CO-Hb was formed in the culture medium when oxy-Hb was incubated with the CO-delivered hepatocytes (Supplementary Fig. S13). We performed an additional control for this experiment, using met-Hb instead of oxy-Hb. Since met-Hb does not bind CO, cells treated with CORM401-E followed by replacement with medium containing met-Hb should still show high amounts of intracellular CO. In fact, CO content in hepatocytes exposed to the CO releaser followed by met-Hb was very similar to that of cells exposed to CORM401-E alone (Supplementary Fig. S12). These results indicate that CO initially accumulated in cells was subsequently transferred to Hb. These results suggest that Hb is capable of

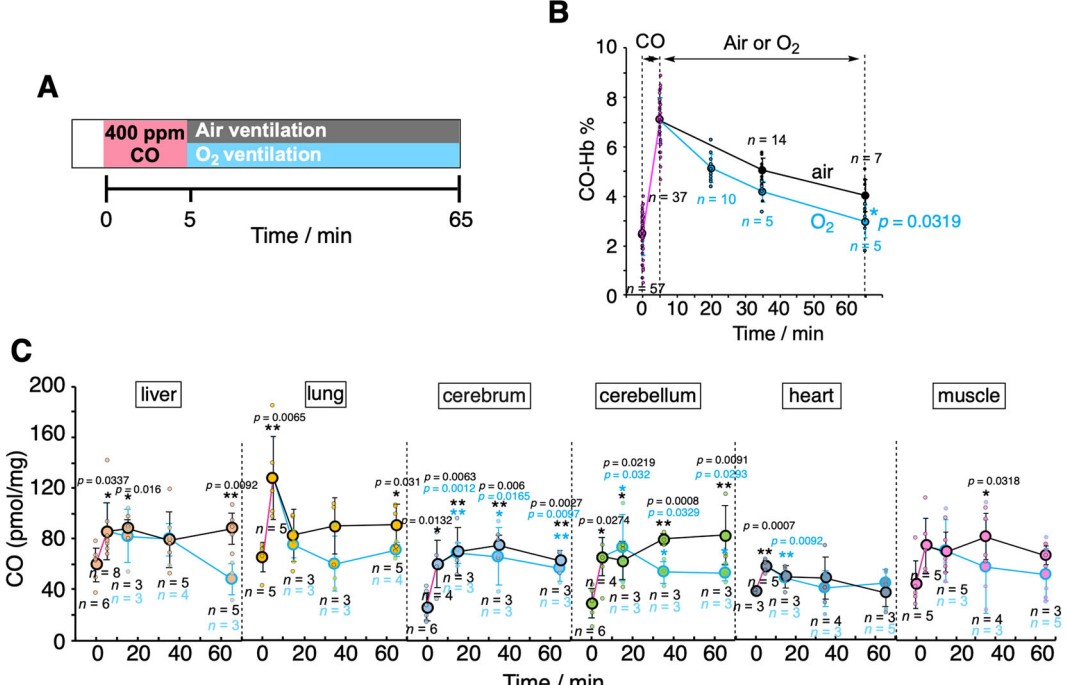

**Fig. 6 Effect of normobaric air/O₂ ventilation on CO levels in blood and tissues after exposure to CO in rats. A** Anesthetized rats were exposed to CO inhalation (400 ppm) for 5 min followed by either air (black) or O₂ ventilation (blue) as indicated. **B** Changes in CO-Hb (%) in the venous blood collected from right ventricles as a function of time. Each plot represents mean ± SD ($n > 5$, the number of experiments is shown in the panel). Statistical significance, $*p < 0.05$, versus air ventilation. **C** Amounts of CO measured under the experimental conditions described in **A**. The plots connected by black and blue lines represent the data obtained under air and O₂ ventilation, respectively. Each bar represents mean ± SD ($n > 3$, the number of experiments is shown in the panel). Statistical significance, $*p < 0.05$, $**p < 0.01$; n.s. not significant versus $t = 0$.

removing CO stored in cells, which was confirmed by the CO quantification assay using hemoCD1, and that this mechanism could also occur in vivo, supporting a possible protective role of Hb against CO intoxication (see "Discussion" for detail).

Next, the kinetic profiles of CO levels in blood and tissues were studied in rats first exposed to 400 ppm CO for 5 min followed by ventilation with air or pure O₂ for 60 min (Fig. 6A). As expected, CO-Hb levels in blood gradually decreased after the two treatments, with O₂ ventilation being more effective than air (Fig. 6B). However, a significant amount of CO still remained in the brain (cerebrum and cerebellum) even after ventilation with air and O₂ (Fig. 6C). Thus, once CO is accumulated in the brain it is not easily eliminated by standard treatments used against CO intoxication.

**Testing hemoCD1 as an antidote against CO intoxication.** Finally, we tested whether oxy-hemoCD1 could act as an effective CO-removal agent in vivo during CO intoxication (Fig. 7). At first, three types of experimental protocols were used as shown in Fig. 7A. In protocols **I** and **II**, rats were subjected to 5 min CO inhalation followed by an intravenous (i.v.) injection of oxy-hemoCD1 during air (**I**) or O₂ ventilation (**II**). In the third protocol (**III**), CO inhalation was followed by O₂ ventilation and, 30 min later, by injection of oxy-hemoCD1. Immediately after the injection, oxy-hemoCD1 was started to be excreted in the urine in the form of CO-hemoCD1 (Supplementary Fig. S14) and the pharmacokinetic study on oxy-hemoCD1 has been reported elsewhere[27,36]. The data in Fig. 7A show that oxy-hemoCD1 infusion fastens the return of CO-Hb to basal levels. Additionally, the infusion of oxy-hemoCD1 elicited a marked reduction in CO accumulation in the brain (cerebrum and cerebellum, see Fig. 7A) and other tissues (Supplementary Fig. S15). The combination of

oxy-hemoCD1 with O₂ ventilation (**II** and **III**) was especially effective in removing CO from the brain.

To simulate more closely a state of severe CO intoxication, the effect of oxy-hemoCD1 administration was further tested in rats after inhalation of 400 ppm CO for 80 min. Under these conditions, CO-Hb levels reached 30% (Fig. 7B). Unexpectedly, the amount of CO accumulated in tissues was not significantly increased compared to that measured after shorter CO inhalation times (5 and 20 min) (Fig. 7A, B and Supplementary Fig. S16). This observation is consistent with the scenario that a high capacity for CO scavenging of circulating Hb impedes accumulation of excess CO in tissues (see Discussion for detail). Also in this case, the significant amount of residual CO detected in the brain after O₂ ventilation for 60 min (**IV**$_{ctl}$/**V**$_{ctl}$) was effectively reduced by i.v. injection with oxy-hemoCD1 (**IV** and **V**).

## Discussion

In this study, we report on the ability of hemoCD1 to effectively remove CO from tissues following CO exposure in rats. Our data show that inhalation of CO gas rapidly increases CO-Hb and CO accumulation in tissues in vivo. While air or O₂ ventilation is capable of restoring CO-Hb to basal levels, elimination of CO from tissues is hard to achieve, especially in the brain. However, infusion of hemoCD1 during the ventilation treatments efficiently eliminates CO from tissues, uncovering a very useful property of hemoCD1 as a CO removing agent to combat CO intoxication. We further reveal how tissues rapidly reach a plateau in CO content during CO exposure while CO-Hb levels in blood continue to rise over time. This phenomenon raises important questions on the role of Hb as a potential molecular shield that defends tissues against accumulation of dangerous levels of CO. The data on CO content in tissues and organs were generated

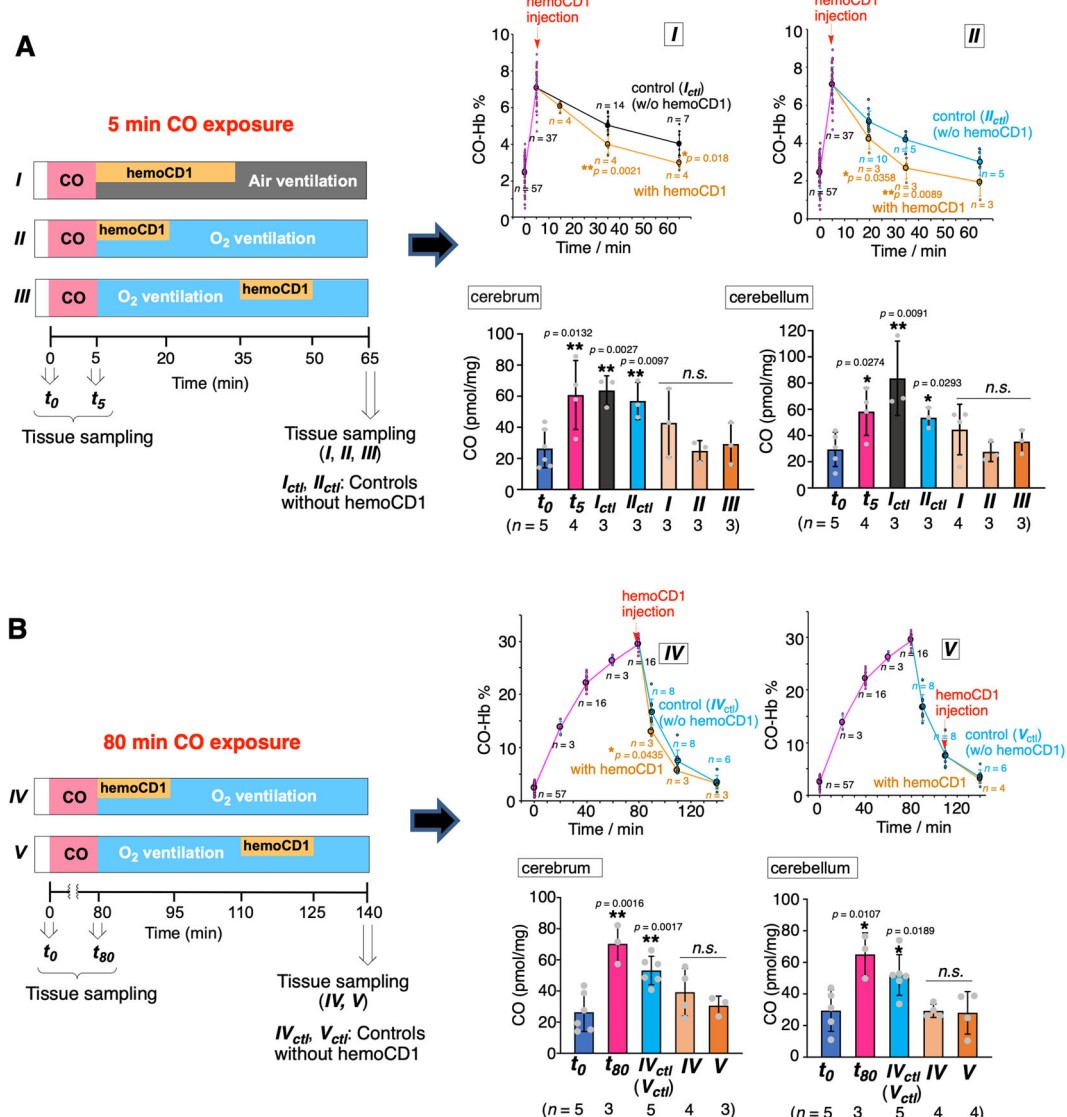

**Fig. 7 Effect of normobaric air/O$_2$ ventilation in combination with hemoCD1 injection on CO levels in blood and tissues after exposure to exogenous CO in rats. A** Anesthetized rats were exposed to CO inhalation (400 ppm) for 5 min followed by either air or pure O$_2$ ventilation in combination with intravenous hemoCD1 infusion (1.4 ± 0.2 mM, 2.5 ml in PBS) as indicated by the three different protocols: **I**, oxy-hemoCD1 was infused for 30 min under room air ventilation; **II**: oxy-hemoCD1 was infused for 15 min under pure O$_2$ ventilation; **III**: O$_2$ ventilation was conducted for 30 min before infusion of oxy-hemoCD1 for 15 min. The right panels show the changes in CO-Hb (%) of the blood and the CO levels detected in the cerebrum and cerebellum samples collected as indicated. **B** Anesthetized rats were exposed to CO inhalation (400 ppm) for 80 min followed by O$_2$ ventilation in combination with intravenous hemoCD1 infusion (3.0 ± 0.2 mM, 2.5 ml in PBS) as indicated by the two different protocols: **IV**, oxy-hemoCD1 was infused for 15 min under pure O$_2$ ventilation; **V**: O$_2$ ventilation was conducted for 30 min before infusion of oxy-hemoCD1 for 15 min. The right panels show the changes in CO-Hb (%) in the blood and the CO levels detected in the cerebrum and cerebellum samples collected as indicated. Each plot for CO-Hb (%) represents mean ± SD ($n > 3$, the numbers of experiments are shown in the panels). Each bar for CO (pmol/mg) represents the mean ± SD ($n > 3$, the numbers of experiments are shown in the panels). Statistical significance, *$p < 0.05$, **$p < 0.01$; n.s. not significant versus $t_0$.

using a new and simplified CO detection method based on hemoCD1, which was thoroughly investigated and validated in the present study.

Our data show that the CO-hemoCD1 complex, unlike CO-Hb, is extremely stable and hardly decomposes under extreme conditions such as high pressures of N$_2$ and O$_2$, or in the presence of oxidants (H$_2$O$_2$). The facts that hemoCD1: (1) is not cell-permeable[39] and not easily denatured by other chemicals; (2) can be isolated from other biocomponents; and that (3) its O$_2$ binding affinity is moderate (Table 2) enabling CO to easily replace O$_2$, make it a unique molecule with suitable properties as a CO scavenging agent in the biological environment. Another

important property is that hemoCD1 also exhibits a much higher CO binding affinity compared to NO and H$_2$S.

Notably, hemoCD1 binds CO with much stronger affinity than Hb, Mb, CcO, and other hemoproteins found in nature (Table 2). The CO binding affinity tends to be higher in Hb-R, neuroglobin (Ngb), CooA, and RcoM; in the case of hemoCD1, the CO affinity is still one order of magnitude higher than that of CO-binding proteins. The kinetic parameters indicate that the high CO binding affinity of hemoCD1 is ascribed to the slow CO off rate ($k_{off}^{CO} = 2.5 \times 10^{-4}\,s^{-1}$). The hydrophobic environment provided by per-O-methyl-β-CD in CO-hemoCD1 tightly holds a hydrophobic CO molecule on the iron(II) center of hemoCD1 as

**Table 2 Kinetic and thermodynamic parameters for $O_2$ and CO bindings of hemoCD1 and related hemoproteins and model compounds[a].**

| | $k_{on}^{CO}$ (M⁻¹s⁻¹) | $k_{off}^{CO}$ (s⁻¹) | $K_d^{CO}$ (M) | $k_{on}^{O2}$ (M⁻¹s⁻¹) | $k_{off}^{O2}$ (s⁻¹) | $K_d^{O2}$ (M) | $K_d^{O2}/K_d^{CO}$ (= M) |
|---|---|---|---|---|---|---|---|
| Hb-R[b,c] | $(4.6\text{–}6.0) \times 10^6$ | $(0.9\text{–}1.9) \times 10^{-2}$ | $(1.7\text{–}4.1) \times 10^{-9}$ | $(3.3\text{–}5.0) \times 10^7$ | 15–22 | $(3.0\text{–}6.7) \times 10^{-7}$ | 150–400 |
| Hb-T[b] | $8.3 \times 10^4$ | $9.0 \times 10^{-2}$ | $1.1 \times 10^{-6}$ | $4.5 \times 10^6$ | $1.9 \times 10^3$ | $4.2 \times 10^{-4}$ | 380 |
| Mb[d] | $5.1 \times 10^5$ | $1.9 \times 10^{-2}$ | $3.7 \times 10^{-8}$ | $1.7 \times 10^7$ | 15 | $8.8 \times 10^{-7}$ | 25 |
| CcO[e] | $(7.0\text{–}12) \times 10^4$ | $2.2 \times 10^{-2}$ | $(1.8\text{–}3.1) \times 10^{-7}$ | $(1.0\text{–}6.0) \times 10^8$ | 10 | $(1.7\text{–}10) \times 10^{-8}$ | 0.1–0.6 |
| Cyt P450scc[f] | $2.2 \times 10^4$ | 0.15 | $7.0 \times 10^{-7}$ | $5.3 \times 10^6$ | 120 | $2.3 \times 10^{-5}$ | 33 |
| NPAS2 (PASA domain)[g] | $3.7 \times 10^5$ | 0.37–0.74 | $(1\text{–}2) \times 10^{-6}$ | – | – | – | – |
| hNgb[h] | $6.5 \times 10^7$ | $1.4 \times 10^{-2}$ | $2.2 \times 10^{-10}$ | $2.5 \times 10^8$ | 0.8 | $3.2 \times 10^{-9}$ | 15 |
| CooA$_{5c\text{-heme}}$[i] | $3.2 \times 10^7$ | 0.02 | $6.3 \times 10^{-10}$ | – | – | – | – |
| RcoM-2[j] | $>10^4$ | $<10^{-6}$ | $<10^{-10}$ | – | – | – | – |
| Mb$_{H64G}$[d] | $5.8 \times 10^6$ | $3.8 \times 10^{-2}$ | $6.6 \times 10^{-9}$ | $1.4 \times 10^8$ | $1.6 \times 10^3$ | $1.1 \times 10^{-5}$ | $1.3 \times 10^3$ |
| Mb$_{H64L}$[d] | $2.6 \times 10^7$ | $2.4 \times 10^{-2}$ | $9.1 \times 10^{-10}$ | $9.8 \times 10^7$ | $4.1 \times 10^3$ | $4.3 \times 10^{-5}$ | $4.8 \times 10^4$ |
| FePiv3 5Clm[k] | $3.6 \times 10^7$ | $7.8 \times 10^{-3}$ | $2.2 \times 10^{-10}$ | $4.3 \times 10^8$ | $2.9 \times 10^3$ | $6.7 \times 10^{-6}$ | $2.7 \times 10^4$ |
| hNgb$_{H64Q\text{-CCC}}$[h] | $1.6 \times 10^8$ | $4.2 \times 10^{-4}$ | $2.6 \times 10^{-12}$ | $7.2 \times 10^8$ | 18 | $2.5 \times 10^{-8}$ | $9.7 \times 10^3$ |
| hemoCD1 | $1.3 \times 10^7$ | $2.5 \times 10^{-4}$ | $1.9 \times 10^{-11}$ | $4.7 \times 10^7$ | 800 | $1.7 \times 10^{-5}$ | $8.9 \times 10^5$ |

[a]These parameters were determined in aqueous solutions at ambient temperatures (20–25 °C), except for FePiv3Cim, which parameters were determined in absolute toluene.
[b]Ref. 43.
[c]Ref. 24.
[d]Ref. 52.
[e]Ref. 68.
[f]Ref. 70.
[g]Ref. 71.
[h]Ref. 63.
[i]Ref. 72, where the CO-binding site in CooA is blocked by a distal proline residue in the native form (6-coordinated heme), thus the actual CO-binding affinity of CooA is smaller ($K_d^{CO}$ ~10⁻⁶ M).
[j]Ref. 73.
[k]Ref. 24,53, where FePiv35Cim is a picket-fence porphyrin whose iron center is intramolecularly coordinated by an imidazole.

discussed elsewhere[30]. In native Hb and Mb, $O_2$/CO binding sites are surrounded by a steric and polar amino acid residue, called distal His, which reduce the CO binding affinities relative to $O_2$[24,43]. Distal mutants such as H64G and H64L Mb showed much higher CO affinity than that in native Mb, resulting in large $M$ values (see Table 2)[52]. In synthetic iron(II)porphyrins such as hemoCD1 and FePiv$_3$5Cim[53], which have no distal functional groups, the CO binding affinity is much higher than that of Mb and Hb[24], although the lipophilic model cannot be used as CO-scavengers in biological media.

In the present study, we refined and optimized an assay using hemoCD1 for the specific detection of CO in tissues and organs. We previously measured CO in cells using oxy-hemoCD1[41], which can detect CO via ligand exchange reaction from $O_2$ to CO but necessitated three troublesome processes: (1) a gel filtration step to remove excess reductant ($Na_2S_2O_4$) for the preparation of oxy-hemoCD1; (2) an ultrafiltration process to separate hemoCD1 from biological contaminants; and (3) a CO gas bubbling step to convert all the hemoCD1 forms to CO-hemoCD1 for determining the final concentration of hemoCD1 ($C_{total}$) in the sample. Here, we have improved our method by eliminating the gel filtration step since the amount of CO in tissues is determined based on the absorbance ratio at 422 and 434 nm in the presence of excess dithionite. Dithionite does not affect the CO scavenging properties of hemoCD1 and facilitates the precipitation of biocomponents present in tissue samples, allowing to obtain clear solutions for absorbance measurements after centrifugation. The ultrafiltration process was thus also omitted. Furthermore, the ratiometric assay using $A_{422}/A_{434}$ and $A_{427}$ enables to determine $M_{CO}$ in a single spectrum measurement, eliminating the final CO bubbling step. We also confirmed that the new assay can be used for cultured cells in vitro, measuring the same amount of endogenous CO as previously reported[41]. Therefore, the new assay using hemoCD1 is accurate and represents a convenient method to determine the amount of CO stored in biological samples such as cells/tissues/organs. We note that different methods to detect CO in tissues and cells have been developed by several groups[10,18–21,54–62]. CO detection in cells has been first achieved with the fluorescent probe COP-1[54] and subsequently other Pd-based CO sensitive probes were

synthesized[56–59] but none of them can quantify CO in samples. Although laser spectroscopy and radio-isotope methods have been reported[60–62], GC remains the most accessible and common technique to quantify CO. As we demonstrated in this study, CO quantification in tissues was underestimated by GC analysis compared to our method, highlighting once more the sensitivity of the hemoCD1 assay. A previous investigation on CO biodistribution measured by GC[21] showed that CO was not accumulated in the brain and muscle tissues when animals were exposed to air containing 2500 ppm CO for 45 min. Although this different trend might be attributed in part to the quantification methods (GC versus hemoCD1 assay), we suggest that the difference in the way animals were exposed to CO is a major factor. In fact, in our experiments 400 ppm CO gas was directly introduced into rats via intubation and under these conditions we measured CO accumulation in brain and muscle within 5 min of exposure. This is an interesting observation showing that the distribution of CO in the tissues differs depending on how CO is inhaled.

Our study on the kinetic profiles of CO in tissues and blood reveals interesting dynamics of CO distribution. First, the amount of CO stored in tissues quickly reached a plateau during CO inhalation but the ex vivo experiments revealed that tissue samples can store more CO. Secondly, despite the plateau observed in tissues, CO-Hb in blood linearly increased during CO exposure. What kind of mechanism could explain this dynamic? We postulate a scenario according to which CO reaching the tissues would gradually transfer to Hb in RBC where it would be stored as CO-Hb before elimination through the lungs. The schematic representation of uptake and elimination of CO is proposed in Fig. 8. In normal conditions, endogenous CO is continuously produced in every cell, some of which is stored in tissues and gradually eliminated through RBC (Fig. 8A). Following inhalation, CO distributes in the body: 95% readily binds to Hb in blood[11] and the rest that is not captured by RBC diffuses to tissues (Fig. 8B). As the CO binding affinity of Hb is much higher than those of Mb and CcO (Table 2), which are the major CO storage components in tissues, it is reasonable to assume that CO flows from tissues to Hb in RBC. Thus, continuous CO inhalation leads to a further increase in CO-Hb, while CO in tissues reaches

**A) Normal conditions**

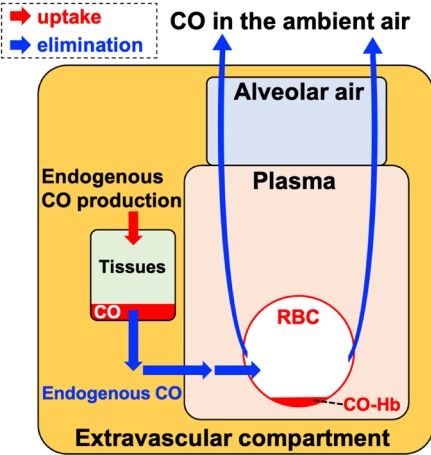

**B) CO inhalation; initial stage (5 min)** **C) CO inhalation; a steady state**

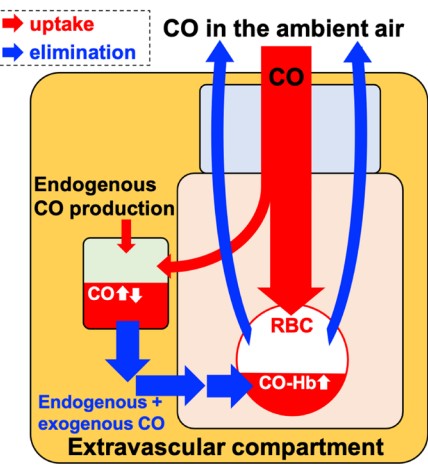

**Fig. 8 Proposed mechanism of CO compartmentalization under normal conditions and during CO inhalation. A** Normal conditions. Endogenous CO continuously produced in cells is stored in tissues, diffuses to Hb, and is exhaled. **B** Initial stage of CO inhalation. Inhaled CO forms CO-Hb in RBC and diffused to tissues. **C** A steady state during CO inhalation. CO accumulated in tissues gradually transfers to Hb in RBC based on the higher CO affinity of Hb versus intracellular CO targets (see Table 1 and text for details). The compartment models are based on our data and Refs. [9,19].

a certain plateau, i.e. a steady state (Fig. 8C). This is corroborated by our two CO exposure protocols (5 and 80 min), since the amount of CO accumulated in tissues was very similar but CO-Hb reached levels of 10% and 30% after inhalation of CO for 5 and 80 min, respectively. The CO flow mechanism suggests that Hb in RBC plays a role in protecting surrounding tissues from CO toxicity. In fact, as demonstrated in the dog study[16] mentioned in the Introduction, high CO-Hb itself is not toxic to the animals and several studies highlight that it is the CO diffused to the tissues and not the fraction bound to Hb that is the principal cause of CO poisoning[8,16,63]. Our data showing that exogenous Hb can capture CO previously accumulated in hepatocytes in vitro support this proposed mechanism. Likewise, the fact that CO accumulation in organs ex vivo is higher than that measured after CO exposure in vivo indicates that, in the absence of Hb, more CO can reach the tissue.

Our results also show that $O_2$ ventilations after CO inhalation is more efficient than air in removing CO from blood. However, significant amounts of CO still remained in the brain tissues even after 60 min $O_2$ ventilation. This different behavior of the brain compared to other tissues might be due to a slower diffusion of

CO across the brain/blood barrier and/or due to the existence of specific proteins with high CO affinity such as Ngb. The residual CO in the brain might be the cause of CO-associated delayed neurological sequelae (DNS)[8,64] as up to 40% of subjects surviving acute CO poisoning may develop DNS in 2–40 days. This may lead to memory loss, movement disorders, and Parkinson-like syndrome among others[64]. The pathophysiology of CO poisoning and subsequent DNS is poorly understood, although CO-induced mitochondrial dysfunction is likely a cause of brain injury[8,65–67]. Our data on CO quantification indicate that CO, once stored in the brain, is more difficult to eliminate than from other tissues. This observation supports the idea that extended $O_2$ ventilation, beyond the normalization of CO-Hb levels, may be necessary to completely remove CO accumulated in brain after intoxication.

Thus, in the last part of our study we tested the potential use of hemoCD1 as an antidote for CO intoxication considering that $O_2$ ventilation is the only method practically used as a therapy. Recently, a neuroglobin mutant (Ngb-H64Q-CCC)[63,68], which showed much higher CO binding affinity than Hb and hemoCD1 (Table 2), has been proposed as an injectable type of CO antidote.

Our data demonstrate that i.v. injection of oxy-hemoCD1 was effective in decreasing CO levels in the brain of CO-treated rats. Since our previous study[40] showed no evidence that hemoCD1 diffuses through the blood brain barrier, we speculate that an equilibrium diffusion of CO to hemoCD1 in plasma effectively reduces CO in the brain. In any case, oxy-hemoCD1 injection reduced CO-Hb in blood[38], which may boost the elimination of CO in tissues by circulating Hb. To investigate the practicality of using hemoCD1 as an injectable CO antidote, we plan to expose animals to increasing CO levels and test for organs and tissue damage. In addition, toxicological studies evaluating hemoCD1 per se will be performed in the near future.

In conclusion, we propose the use of hemoCD1 as a useful CO scavenger for quantifying and removing exogenous CO in tissues. Our study emphasizes: (1) the high affinity of hemoCD1 for CO and the chemical stability of the CO-hemoCD1 complex, demonstrating the suitability of hemoCD1 as a sensitive CO scavenger; (2) that the ratiometric absorbance assay using hemoCD1 in the presence of dithionite is a very simple and sensitive method for quantifying CO contained in tissues; (3) an unrecognized role of Hb as a predominant CO scavenging system in vivo, preventing accumulation of excess CO in tissues; (4) that injection of oxy-hemoCD1 to CO-exposed rats acts as adjuvant to air/$O_2$ ventilation to effectively and rapidly remove CO accumulated in tissues, including the brain. We believe that the data presented herein will stimulate further discussions on the mechanisms underlying CO poisoning.

## Methods

**Preparation of ferrous hemoCD1**. 5,10,15,20-Tetrakis(4-sulfonatopheyl)porphinatoiron(III) ($Fe^{III}TPPS$) and Py3CD were synthesized in our laboratory[29,30]. Stock solutions of hemoCD1 used for CO quantification in tissues were prepared as follows. $Fe^{III}TPPS$ (1.10 mg, 1.0 μmol) and Py3CD (3.53 mg, 1.2 μmol) were dissolved in PBS (1 ml) to yield a solution of met-hemoCD1 (1 mM). The stock solution of met-hemoCD1 (1–5 μl) was appropriately diluted with PBS to prepare the solution of met-hemoCD1 (2–10 μM, 0.5 ml) for CO quantification in tissues. Ferrous deoxy-hemoCD1 was obtained by adding an excess of $Na_2S_2O_4$ (ca. 1–2 mg) to the met-hemoCD1 solution (2–10 μM, 0.5 ml).

The solution of oxy-hemoCD1 used as a CO removal agent to be injected in rats was prepared as follows. $Fe^{III}TPPS$ (6.60 mg, 6.0 μmol) and Py3CD (21.15 mg, 7.2 μmol) were dissolved in PBS (2 ml) to yield a solution of met-hemoCD1 (3 mM). $Na_2S_2O_4$ (ca. 10–20 mg) was then added to reduce met-hemoCD1 to ferrous deoxy-hemoCD1. Excess $Na_2S_2O_4$ was removed by passing the deoxy-hemoCD1 solution through a HiTrap desalting column (Sephadex G25, GE Healthcare Life Sciences). During the filtration process, deoxy-hemoCD1 was converted to oxy-hemoCD1 by capturing atmospheric $O_2$. The concentration of oxy-hemoCD1 was determined from its absorption coefficient $\varepsilon_{422}^{oxy} = 1.64 \times 10^5 \, M^{-1} \, cm^{-1}$ [33,41].

**Animals**. Five weeks old Lewis and Sprague-Dawley rats were purchased from Charles River Co. Ltd. (Yokohama, Japan). Before undergoing any experiment, animals were acclimatized for a week under air conditioning at 26 ± 0.5 °C with access to water and food ad libitum.

**Animal preparation**. All experiments were approved by the Institutional Review Board of Tokai University and the Guidelines for Animal Experiments of Doshisha University. Animals received humane care as required by the institutional guidelines for animal care and treatment in experimental investigations according to the Guide for the Care and Use of Laboratory Animals (Institute of Laboratory Animal Resources, 1996). Rats were anesthetized with 3% sevoflurane, orally intubated and ventilated at 15 ml/kg of tidal volume with ambient air at a ventilation rate of 55 per min with no end-expiratory pressure by a ventilator (Rodent Ventilator, Ugo Basile). An indwelling catheter was placed in the tail vein and saline was administered at 3 ml/h. The body temperature was monitored with a rectal probe and maintained at 36.5 ± 0.5 °C with a water blanket (MEDI-Therm II, Gaymer Industries Inc.). The chest was opened at the 4th intercostal space, the dose of sevoflurane was reduced to 2%, and rats were anesthetized with an intraperitoneal injection of pentobarbital (PB). After 5 min, sevoflurane anesthesia was removed and rats were ready for the experiments with CO gas inhalation (see below).

**Blood CO-Hb measurements**. Rat venous blood samples (0.1 ml) were regularly collected from right ventricles. Blood samples were immediately analyzed using a blood gas analyzer, ABL825 (Radiometer Co. Ltd.), which measures CO-Hb based on a 128 wavelengths spectrometer with a measuring range from 478 to 672 nm.

**Preparation of tissue samples**. Tissue specimens from liver, lung, cerebrum, cerebellum, heart (myocardium), and skeletal muscle were collected immediately after the chest was opened. To remove blood from the organs, saline was flushed before collecting the tissues as follows: the blood circulation was perfused with saline (50 ml) through the pulmonary artery and then puncturing the left ventricle with a 16G needle (TERUMO). Additional saline (150 ml) was then injected at 0.25 ml/s using a peristaltic pump. The flushed tissues were collected and immediately frozen in liquid nitrogen and stored at –80 °C prior to CO measurements.

**Quantification of CO in tissues by the hemoCD1 assay**. Tissue samples (5–20 mg) harvested from rat organs were weighed and homogenized by Power MasherII (nippi) in PBS (0.5 ml). After homogenization, deoxy-hemoCD1 (2–10 μM) with $Na_2S_2O_4$ (ca. 1–2 mg) in PBS (0.5 ml) was added to the tissue homogenates, which were disrupted by sonication (time: 10 s × 2, on ice, amplitude: 15; QSONICA). Samples were centrifuged (14,000 × g, 15 min) and the supernatants filtered (DISMIC 13CP, 0.45 μm pore; ADVANTEC). The filtrates were treated with $Na_2S_2O_4$ (ca. 1–2 mg) before measurements by UV–vis absorption spectroscopy (NanoPhotometer® C40, Implen). The concentration of total hemoCD1 ($C_{total}$) was determined from the absorbance at 427 nm by Eq. (2) as follows:

$$A_{427 \, nm} = \varepsilon_{427} \cdot C_{total} \cdot l \tag{2}$$

where $\varepsilon_{427}$ is the molar extinction coefficient at 427 nm ($1.95 \times 10^5 \, M^{-1} \, cm^{-1}$), at which the wavelength of the isosbestic point of deoxy- and CO-hemoCD1. The $l$ is the optical path length (1.0 cm). The resulting hemoCD1 solution contains CO-hemoCD1 ($C_{co}$) and deoxy-hemoCD1 ($C_{deoxy}$) (Eq. (3)):

$$C_{total} = C_{co} + C_{deoxy} \tag{3}$$

The ratio of CO-hemoCD1 in total hemoCD1 ($R_{CO}$) is represented as Eq. (4):

$$R_{CO} = C_{co}/C_{total} \tag{4}$$

The absorbances at 422 nm and 434 nm are defined as Eqs. (5) and (6):

$$A_{422} = \varepsilon_{deoxy}^{422} \cdot C_{deoxy} + \varepsilon_{co}^{422} \cdot C_{co} \tag{5}$$

$$A_{434} = \varepsilon_{deoxy}^{434} \cdot C_{deoxy} + \varepsilon_{co}^{434} \cdot C_{co} \tag{6}$$

Therefore, the ratio of $A_{422}/A_{434}$ is represented as Eq. (7):

$$\frac{A_{422}}{A_{434}} = \frac{\varepsilon_{deoxy}^{422} \cdot C_{deoxy} + \varepsilon_{CO}^{422} \cdot C_{CO}}{\varepsilon_{deoxy}^{434} \cdot C_{deoxy} + \varepsilon_{CO}^{434} \cdot C_{CO}} \tag{7}$$

From Eqs. (3)–(7), $R_{CO}$ is represented as Eq. (8):

$$R^{CO} = \frac{\varepsilon_{deoxy}^{422} - A_{422}/A_{434} \cdot \varepsilon_{deoxy}^{434}}{A_{422}/A_{434}(\varepsilon_{CO}^{434} - \varepsilon_{deoxy}^{434}) - \varepsilon_{CO}^{422} + \varepsilon_{deoxy}^{422}} \tag{8}$$

where $\varepsilon_{co}^{422} = 3.71 \times 10^5 \, M^{-1} \, cm^{-1}$, $\varepsilon_{co}^{434} = 6.75 \times 10^4 \, M^{-1} \, cm^{-1}$, $\varepsilon_{deoxy}^{422} = 1.52 \times 10^5 \, M^{-1} \, cm^{-1}$, and $\varepsilon_{deoxy}^{434} = 2.13 \times 10^5 \, M^{-1} \, cm^{-1}$ are used for calculation. The moles of CO contained as CO-hemoCD1 in solution ($M_{CO}$) are determined by Eq. (9):

$$M_{CO}(mol) = R_{CO} \cdot C_{total} \cdot V \tag{9}$$

where $V$ is the volume of the solution ($1 \times 10^{-3}$ l). The relation between the $R_{CO}$ and $A_{422}/A_{434}$ values in Eq. (8) is nonlinear, and thus reproducibility of $R_{CO}$ would become low when $A_{422}/A_{434}$ below 0.8 ($R_{CO} = 0.055$) or over 3.0 ($R_{CO} = 0.743$). For accurate quantification, the $A_{422}/A_{434}$ values were thus adjusted between 0.8 and 3.0 by slightly varying the initial amount of tissues or hemoCD1 added to the samples.

**Protocol for exposure of tissues to CO ex vivo**. Tissue samples were placed in a 5 ml 25G syringe (TERUMO) sealed with a septum cap (natural rubber, for a 7 mm tube). The atmosphere in the syringe was replaced with pure CO gas (4 ml) (Sumitomo Seika) through inserting a 25G needle (TERUMO) into the septum cap. Tissue samples were then incubated at 4 °C under a CO atmosphere. After 1 h, the syringe was opened under air atmosphere, samples were taken out from the syringe, and the amount of CO was measured by the hemoCD1 assay.

**Quantification of CO in tissues by gas chromatography**. The amount of CO in the tissue samples was measured by GC according to the reported method[18] with slight modifications. Tissues (100 mg) were homogenized (Power MasherII, nippi) in PBS (0.3 ml) and then disrupted by sonication (time: 10 s × 2, on ice, amplitude: 15; QSONICA). The solution and a 5 mmφ glass ball were placed in a 5 mL 25 G syringe (TERUMO) sealed with a septum cap (natural rubber, for a 7 mm tube). The atmosphere in the syringe (1 mL) was carefully replaced with He gas through inserting a 25 G needle (TERUMO) into the septum cap. Three drops of 30% sulfosalicylic acid (SSA) (Wako) were injected through the septum into the syringe. The mixture was then mixed well using a 5 mmφ glass ball. Methane gas (50 μl) was added to the head-space atmosphere through the septum in the syringe as an internal standard. CO liberated from the tissues into the gas phase (0.5 ml) was analyzed by GC (Shimadzu GC-2014, Shimadzu). The GC conditions were as follows: detector = TCD; column = SHINCARBON: carrier gas = He; column

temperature = 40 °C; temperature at vaporizing chamber and detector = 120 °C; flow rate = 50 ml/min.

**Exposure to CO gas inhalation in vivo**. The experimental setup for CO gas inhalation in vivo is shown in Fig. S11. After anesthesia by an intraperitoneal (i.p.) injection of PB, rats were exposed to 400 ppm CO gas equilibrated in air (flow: 1 l/min, TOMOE SHOKAI Co., LTD.) through oral intubation and ventilation (Rodent Ventilator, Ugo Basile) for 5, 10, or 20 min. The concentration of the inhaled CO gas was monitored by a CO detector (COSMOS XC2200). The exhaled gas was diluted with air before exiting and the atmosphere was monitored during the entire experiment. During CO exposure, venous blood from right ventricles (RV, 0.1 ml each) were collected and analyzed using the blood gas analyzer as described above. At each time point, organs were flushed as described above and tissue samples collected and frozen were finally stored at –80 °C for further CO measurements.

**Protocols of air/O$_2$ ventilations after exposure to CO in vivo**. The experimental setup was the same as shown in Fig. S11. Anesthetized rats were exposed to CO gas (400 ppm) for 5 min as described above, and then breathing gas was changed to either air or pure O$_2$ (TOMOE SHOKAI Co., LTD.). Venous blood samples (0.1 ml) were collected for blood gas analyses and tissues were harvested at different time points. Tissues were frozen in liquid nitrogen and stored at –80 °C for further CO measurements.

**Administration of oxy-hemoCD1 to rats exposed to CO**. The solution of oxy-hemoCD1 (1.4–3.0 mM, 2.5 ml in PBS) was freshly prepared before each experiment to avoid autoxidation. Before the experiments, we confirmed that in the blood gas analysis (ABL825, Radiometer Co. Ltd.) at 128 different wavelengths the Hb spectrum was unaffected by the oxy-hemoCD1 administration because the injected amount of oxy-hemoCD1 was much less (ca. 1/100) than Hb in blood.

In the case of the experiments conducted according to protocol *I*, the solution of oxy-hemoCD1 (1.4–3.0 mM, 2.5 ml in PBS) was first prepared in a syringe and placed in a syringe pump (TERUMO) ready to be injected. Five minutes after exposure of rats to CO gas (400 ppm), the inhaled gas was switched to air and 1 ml of oxy-hemoCD1 was quickly infused from the tail vein. The rest of the solution of oxy-hemoCD1 (1.5 ml) was then infused through the same vein at a rate of 4.5 ml/h (the total infusion time: 30 min). At the end of infusion, rats were kept on air ventilation for an additional 30 min. Venous blood samples were collected at 0, 5, 15, 35, and 65 min time points ($t_0$, $t_5$, $t_{15}$, $t_{35}$, $t_{65}$). At the same time points, the urine was collected and analyzed by UV–vis absorption spectroscopy (NanoPhotometer® C40, Implen). At each time point, the chest was opened, tissues were flushed using saline (200 ml), collected and frozen in liquid nitrogen. Samples were finally stored at –80 °C for further CO measurements.

In the case of the experiments carried out according to protocol *II*, the procedure was the same as in protocol *I* except that after exposure of rats to CO, the inhaled gas was switched to pure O$_2$ and 1 ml of oxy-hemoCD1 was quickly infused intravenously. The rest of the solution of oxy-hemoCD1 (1.5 ml) was then infused in the same vein at a rate of 9.0 ml/h (the total infusion time: 15 min). After the end of infusion, the rats were kept on pure O$_2$ ventilation for an additional 45 min. Venous blood samples and tissues were collected using the same procedures as in protocol *I*.

In the case of the experiments conducted according to protocol *III*, rats were treated with CO gas as in the other protocols, then the gas was switched to pure O$_2$. After 30 min, a solution of oxy-hemoCD1 (1 ml) was quickly infused from the tail vein and the rest (1.5 ml) was infused to the same vein at a rate of 9.0 ml/h (the total infusion time: 15 min). At the end of infusion, rats were kept on O$_2$ ventilation for an additional 15 min. Venous blood samples and tissues were collected using the same procedures as in protocol *I*.

In the case of the experiments conducted according to protocols *IV* and *V* (Fig. 7B), rats were treated with 400 ppm CO gas for 80 min, then the gas was switched to pure O$_2$. In case of 80 min inhalation, 3.0 mM oxy-hemoCD solution (2.5 ml) was used for intravenous injection. Subsequent procedures were the same as in protocols *II* and *III*, respectively.

**Cell cultures**. The human hepatoma cell line (HepG2) was obtained from RIKEN Cell Bank and cultured in Dulbecco's modified Eagle's medium (DMEM, GIBCO) supplemented with 10% fetal bovine serum (Invitrogen GIBCO, heat inactivated at 56 °C before use) and 1% penicillin/streptomycin/amphotericin B (Wako) at 37 °C in a humidified atmosphere in the presence of 5% CO$_2$. Human met-Hb was purchased from Sigma-Aldrich. Ferrous Hb was prepared using Na$_2$S$_2$O$_4$ as a reducing agent, followed by purification using a HiTrap desalting column, Sephadex G25, GE Healthcare Life Sciences. The concentration of Hb was determined using reported absorption coefficients[69].

**In vitro experiments to demonstrate CO transfer from cells to Hb**. HepG2 (1 × 10$^6$) cells were treated with CORM401-E (25 µM in D-10) for 2 h at 37 °C. The medium was then removed, cells were washed with PBS, and then incubated with oxy-Hb (15 µM in D-10) for 1 h. Cells were washed with PBS, collected by scraping and the suspension was sonicated. The amounts of CO contained in the cells were then quantified by the hemoCD1 assay.

**Effect of hemoCD1 and Py3CD on cell viability**. Cell viability was assessed by the MTT assay. Briefly, HepG2 cells (ca. 10$^5$ cells) in D-10 (100 µl) were seeded into each well of 96-well plates and incubated at 37 °C in a humidified CO$_2$ atmosphere (5%) for 24 h. The medium was replaced with fresh medium (D-10) containing Py3CD (20, 50, 100, 150, and 250 µM) or met-hemoCD1 (20, 50, 100, 150, and 250 µM) and treated for 3 h. MTT (Sigma) was dissolved in PBS at a concentration of 5 mg/ml. After 3 h, 100 µl of medium was removed and 10 µl of the MTT solution with 90 µl D-10 was added to each well. After 3 h incubation at 37 °C, 100 µl of medium was removed and mixed with 100 µl DMSO (Wako) before absorbance at 570 nm was determined on a Multiskan™ FC Microplate Photometer (Thermo Fisher Scientific).

**Statistics and reproducibility**. All data represent the means ± standard error from at least three different experiments and were analyzed by unpaired Student's *t* tests. Differences with *p* values of <0.05 were considered significant. For experiments involving quantification of CO, *n* = 3 was chosen as the minimal replicate number. No data were excluded. All replication experiments were confirmed successful. The tissue samples from rats were taken and selected randomly in the measurements. The collection and analysis of the tissue samples were independently performed, and a partly blinded analysis was conducted.

**Reporting summary**. Further information on research design is available in the Nature Research Reporting Summary linked to this article.

## Data availability
The authors declare that the data supporting the findings of this study are available within the paper and the Supplementary information files. Source data for charts in the main figures is available in Supplementary Data 1 and 2.

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

## Acknowledgements

This work was financially supported by MEXT/JSPS KAKENHI (Grant Nos. JP15H02569, JP17H02208, JP18KK0156, JP19K22260, JP19K22972, JP20H02871), the MEXT-Supported Program for the Strategic Research Foundation at Private Universities (2015–2019), the Takeda Science Foundation, the NOVARTIS Foundation (Japan) for the Promotion of Science, the Suntory Foundation for Life Sciences, and the JGC-S Scholarship Foundation. Q.M. deeply thanks the support of Otsuka Toshimi Scholarship Foundation and a Visiting Fellowship from IMRB/Inserm. R.F. and R.M. are supported by a grant from the Agence National de la Recherche (ANR-19-CE18-003201SWEET-CO).

## Author contributions

Q.M., A.T.K., S.M., and H.K. performed experiments and analyzed data; Q.M., R.M., R.F., and H.K. conceived the concept of the study and wrote the manuscript.

## Competing interests

The authors declare no competing interests.
