## [Peer Review File · Communications Biology]

Reviewers' comments:

Reviewer #1 (Remarks to the Author):

The manuscript by Mao et al. reports on a improved test to measure CO in biological samples, from cell culture, tissues etc. CO is an important endogenous molecules but also a toxic gas, when exposed to too high levels. The manuscript is very complete, very clear to follow and the results are well presented. Although I am familiar with bioinorganic chemistry in general, heme and CO are not my expertise. Thus I appreciated this clarity a lot. The research was well put into context and the improvements in the new test compared to their last published test are analyzed. The new test is simpler (no SEC and ultrafiltration) and the large panel of results to show the robustness of the test. All this justifies publication.

Experiments were repeated several times, at least $n=3$, but often more and statistically analyzed. I am not an expert in statistic, but to me it seems well done. May be more importantly, results and conclusions were confirmed by different type of experiments with different type of samples and compared with standards. Moreover, the authors tried to challenge the test (by adding a lot of potential different competitors). This all together makes the test very robust and shows that GC likely underestimates CO content.

Another interesting part of the manuscript is the role of hemoglobin. In the beginnings it was though CO binding to hemoglobin (Hb) is a main toxic mechanism. Here the results suggest a protective role and detoxification role. Hb can act in two ways, one by shielding tissues from higher CO exposure by capturing CO and the other by depleting CO from the tissue due to its CO-binding capacity. Moreover the work shows also that hemoCD1 could be an efficient adjuvant to O₂ ventilation to remove CO from tissues and body.

In summary, the work is interesting and sheds a new light on the role of Hb in CO toxicity. The developed test is very useful (in particular if the molecule is accessible for other researchers). The work is thoroughly done and honestly discussed, features unfortunately less and less common. Thus I recommend strongly publication after the following points have been considered.

- 1) Experiment Fig.S12. In vitro experiments to demonstrate CO transfer from tissues to Hb: Did the authors try to measure the CO on the Hb to see if it ends up to the total CO released (e.g. via Hb-CO spectrum or spectroscopy or with the hemoCD1 test)? This would give direct evidence of CO on Hb, and not only the indirect via the absence of CO in the tissue. Would Met-Hb have been a useful control? I
- 2) Fig. 8. It took me a while to clearly see the message. I was stumbling over two points. First, why are the blue arrows reflecting CO going from tissues to Hb thicker in C compared to B, although the CO content of the tissue is the same? After I realized the small white arrows indicating that CO tissue and Hb-CO are increasing in B. To be consistent the blue arrows from tissue to Hb should also be on the rise, or the level of CO in the tissue should be a little lower than in C. Second, the blue arrows indicating CO leaving the system from Hb-CO have the same width in all (A, B and C). These arrows should become larger in C, as more CO is released. By the way, I think C is better described as a steady state system and not as an equilibrium state. May be the authors find a way to improve the clarity of the figure.
- 3) Please indicate the concentrations used in in the caption for Fig 2A to D.

Peter Faller, Strasbourg

Reviewer #2 (Remarks to the Author):

Mao et al present a very interesting study on the effect of hemoCD1 – a high affinity CO-binding molecule developed by the authors- as a CO scavenger in a rat model. In addition, they use hemoCD1 to assess CO levels in tissues. Based on the experimental results, the authors validate hemoCD1 as a promising tool to estimate cellular and tissue CO levels, and conclude that hemoCD1 is a viable CO removal agent. The authors also propose a role for blood Hb to serve as a protective sink for CO during CO inhalation.

The study design is quite thorough, although on occasion it may be more convoluted than necessary. The conclusions are in general well supported by the data. Overall this is an interesting work and, in my opinion, it deserves publication with some corrections. My specific comments are indicated below-

1. The abstract notes that “It is widely believed that an increase in blood carboxyhemoglobin (CO-Hb) is the best biomarker to define CO intoxication, neglecting the important fact that CO accumulation in tissues is the most likely direct cause of mortality”. I am not sure if “neglecting” is the right word. I think that most physicians working in the CO poisoning field are aware that COHb levels do not necessarily correlate with severity. If mechanisms to assess tissue CO in real time instead of (or probably in conjunction with) circulating CO were available, I am sure that they would be widely used. Also note that one of the conclusions of this work is that tissue CO levels do not tell us much about severity either, so circulating COHb may still be the best available indicator despite its limitations.

2. In the introduction the authors indicate that O₂ ventilation is the major approach for the treatment of CO intoxication and they cite references 12 and 13. Note that these references focus on hyperbaric oxygen (typically 2-3atm) and not the more widely accessible (and used) normobaric (1atm) 100% O₂. I assume that the authors want to include both normobaric and hyperbaric 100%O₂ treatment as they also focus part of their study on normobaric 100% oxygen.

3. Page 8- although the authors conclude that hemoCD1 is safe, there is no toxicology data provided (e.g. markers of heart/liver/kidney damage). The authors refer to previous work (Kitagishi et al (2010) A diatomic molecule receptor that removes CO in a living organism. *Angew. Chem. Int. Ed.* 49, 1312–1315) but the dosage used in that and other previous work was much lower than in the present work and no toxicology report was provided. With the stability data provided I don't foresee major problems, but some kind of in vivo toxicology study would be helpful.

4. In page 10; the authors should note in the main text that the determination requires dithionite to eliminate any interference from the oxidized hemoCD1 species (this is included in other sections but not here).

5. Page 15; as the authors demonstrate that Hb removes CO from the cell, it actually would indicate that HbCO levels are relevant to assess the extent of CO poisoning (or at least more relevant than tissue CO levels)

6. Page 16; when the authors indicate that “Thus, once CO is accumulated in tissues it is not easily eliminated by standard treatments used against CO intoxication.” This seems to be a very specific property of the brain and not so much of other tissues. I would emphasize that point as the sentence in its current form is misleading

7. I am confident that the method proposed by the authors is probably more sensitive than previous accounts; however, there are significant differences in the trends observed by others (Cronje et al (2004). Carbon monoxide actuates O₂-limited heme degradation in the rat brain. Free Radical Biology and Medicine, 37, 1802-1812) and those discrepancies should be discussed in more detail. Note those studies did observe CO accumulation in other organs such as heart but not in brain.

8. In figure 5B and others; the levels in the blood in rest conditions seem way too high. At least in human subjects it is hard to see HbCO levels of 2% or higher in non-smokers. It would be important to validate the HbCO levels with samples quantified by UV-visible spectroscopy

9. In page 18 is noted that "oxy-hemoCD1 infusion during air ventilation (I) fastens the return of CO-Hb to basal levels, while during O₂ ventilation (II) the effect of hemoCD1 infusion is less pronounced." I would note instead that the effects of O₂ and hemoCD1 seem to be additive, which is to be expected due to their mechanisms of action (but good to be experimentally corroborated)

10. Also in page 18 "this... is consistent with the scenario that a high capacity for CO storage of circulating Hb impedes accumulation of excess CO in tissues". It also indicates that the proteins responsible for CO accumulation in tissues – supposedly myoglobin and Cyt c oxidase- have lower CO affinity than Hb. It begs the question of what is the particular issue with the brain. Is this due to slower diffusion of CO across the brain/blood barrier or is this due to specific proteins in the brain with higher CO affinity, e.g. neuroglobin? Other authors have speculated on heme oxygenase activity (Cronje et al (2004). Carbon monoxide actuates O₂-limited heme degradation in the rat brain. Free Radical Biology and Medicine, 37, 1802-1812) but that seems too slow to account for the observed differences in this short timeframe

11. For the model with the 5 min exposure (Figure 7A) it is not clear why the authors change 2 variables in the air vs oxygen tests, as the hemo CD1 infusion time is changes from 30 min to 15 min. I commend the authors for trying the 100% O₂ model, but note that the clearance in rats with room air is already much faster than for humans breathing 100% O₂; so other than the comparison of models in figure 6 it would be easier to monitor the changes in CO clearance comparing the effects in the room air model

12. In page 21 – "The CO binding affinity tends to be higher in CO-sensing proteins such as neuroglobin (Ngb), NPAS2, CoxA, and RcoM". The function of Ngb is unclear and it is also unknown if Ngb is a CO sensing protein at all; also the CO affinity of NPAS2 is not particularly high. From these 4 proteins only CoxA and RcoM are bona fide CO sensors with high CO affinity

13. Page 23. When the authors indicate "We postulate a scenario according to which CO reaching the tissues would gradually transfer to Hb in RBC" this could make sense for the lung, but what would be the route taking CO to the tissues if it is not HbCO as well? It is hard to believe that there is going to be much CO dissolved in plasma as long as there is available Hb around. It would be interesting to estimate such values mathematically

14. Figure 8 is not particularly clear and the left/right duplication in each panel does not help

15. In page 23 – "that it is the CO diffused to the tissues and not the fraction bound to Hb that is the principal cause of CO poisoning" However the steady-state amount diffused seems to plateau. If a longer exposure does not increase the tissue CO levels, it would seem that the relevant variable is the

time that the tissue is subject to the high CO levels. As the authors point out a role of mitochondrial dysfunction on tissue damage – which seems reasonable- perhaps some markers of mitochondrial dysfunction such as GDF15 may provide a better indication of the CO induced damage

16. page 24- I agree with the authors regarding the DNS and it may be helpful to comment on some of the trials using oxygen therapy for longer times, even over several days. Although the results are very heterogeneous, the present data would support extended use of oxygen therapy beyond the normalization of HbCO levels

Reviewer #3 (Remarks to the Author):

Qiyue Mao et al. have reported in this manuscript on highly sensitive quantification of carbon monoxide exploiting a synthetic supramolecular compound composed of an iron(II)porphyrin and a cyclodextrin dimer (hemoCD1) that also can serve as a CO scavenger to eliminate residual CO accumulated in organs. The authors demonstrated that the CO quantification can be achieved by a simple colorimetric assay using absorbances at three different wavelengths, because the complex of CO and hemoCD1 is stable enough for such detection procedure. The high stability of the CO-hemoCD1 complex is extremely high and thus once CO is bound to hemoCD1, CO hardly dissociates. The high stability of CO-hemoCD1 complex allow us to utilize hemoCD1 for kinetic measurements to understand the distribution of CO in organs including brain. Finally, the authors have concluded that a protective role of circulating hemoglobin in CO intoxication. Although excellent results were obtained including highly sensitive quantification of carbon monoxide and one of possible functions of hemoCD1 as a CO scavenger, following questions remained open before publication.

1. The stability of CO-hemoCD1 is extremely high and thus it can be used as a scavenger of CO. Is it possible to release the CO in the physiological conditions? If not, why does hemoCD1 acts as effective adjuvant to O₂ ventilation to eliminate CO?
2. Closely related to the above question, is it possible to selectively remove CO-hemoCD1 (or CO bound to hemoCD1) after injection to rat?
3. Although the results are clear, the importance of circulating Hb as a CO scavenger would be somehow a matter of course. If I would understand results correctly, one of the most important findings would be that it is extremely difficult to remove CO once accumulated in cell even with Hb. Therefore, the title, "Highly sensitive quantification of carbon monoxide (CO) in vivo reveals a protective role of circulating hemoglobin in CO intoxication" would be not suitable.

Reply to Reviewer #1:

We thank the Reviewer for carefully evaluating our manuscript and giving us constructive comments.

1) Experiment Fig.S12. In vitro experiments to demonstrate CO transfer from tissues to Hb: Did the authors try to measure the CO on the Hb to see if it ends up to the total CO released (e.g. via Hb-CO spectrum or spectroscopy or with the hemoCD1 test)? This would give direct evidence of CO on Hb, and not only the indirect via the absence of CO in the tissue. Would Met-Hb have been a useful control?

This is a very interesting set of experiments and we fully agree with the Reviewer that they would be very useful to strengthen the message of our study. We have conducted new experiments by quantifying CO-Hb in the medium and by performing a control experiment using met-Hb, which does not bind CO (see Figure S12 and S13 in the revised manuscript). The results from both sets of experiments support our conclusion that there is transfer of CO from the cell insides to Hb in the medium. We thank the Reviewer for these important suggestions that have helped to clarify this process.

We have now added Figure S13 as a new figure in the revised manuscript (Supplementary section) as well as a new sentence in the Results section, at the end of page 15m as follows: *'Spectroscopic analysis revealed that CO-Hb was formed in the culture medium when oxy-Hb was incubated with the CO-delivered hepatocytes (Fig. S13). We performed an additional control for this experiment, using met-Hb instead of oxy-Hb. Since met-Hb does not bind CO, cells treated with CORM401-E followed by replacement with medium containing met-Hb should still show high amounts of intracellular CO. In fact, CO content in hepatocytes exposed to the CO releaser followed by met-Hb was very similar to that of cells exposed to CORM401-E alone (Fig. S12).'*

2) Fig. 8. It took me a while to clearly see the message. I was stumbling over two points. First, why are the blue arrows reflecting CO going from tissues to Hb thicker in C compared to B, although the CO content of the tissue is the same? After I realized the small white arrows indicating that CO tissue and Hb-CO are increasing in B. To be consistent the blue arrows from tissue to Hb should also be on the rise, or the level of CO in the tissue should be a little lower than in C. Second, the blue arrows indicating CO leaving the system from Hb-CO have the same width in all (A,B and C). These arrows should become larger in C, as more CO is released. By the way, I think C is better described as a steady state system and not as an equilibrium state. May be the authors find a way to improve the clarity of the figure.

We agree with the points raised. The suggested modifications have been introduced in the revised scheme. Additionally, we removed the duplicated part in Figure 8 as Reviewer 2 requested in her/his Q14. We hope Figure 8 is now clear.

3) Please indicate the concentrations used in in the caption for Fig 2A to D.

These concentrations were added in the caption of Figure 2.

Reply to Reviewer #2:

We thank the Reviewer for her/his thoughtful evaluation of our study that helped us to improve our study. Below is our reply to all the points raised.

1. The abstract notes that “It is widely believed that an increase in blood carboxyhemoglobin (CO-Hb) is the best biomarker to define CO intoxication, neglecting the important fact that CO accumulation in tissues is the most likely direct cause of mortality”. I am not sure if “neglecting” is the right word. I think that most physicians working in the CO poisoning field are aware that COHb levels do not necessarily correlate with severity. If mechanisms to assess tissue CO in real time instead of (or probably in conjunction with) circulating CO were available, I am sure that they would be widely used. Also note that one of the conclusions of this work is that tissue CO levels do not tell us much about severity either, so circulating COHb may still be the best available indicator despite its limitations.

We agree that CO-Hb remains at present an essential biomarker to monitor CO intoxication and that physicians do not have other appropriate means to assess the tissue CO levels. Therefore, we accordingly changed the sentence in the abstract as follows: *‘It is widely believed that an increase in blood carboxyhemoglobin (CO-Hb) is the best biomarker to define CO intoxication, while the fact that CO accumulation in tissues is the most likely direct cause of mortality is less investigated.’*

2. In the introduction the authors indicate that O₂ ventilation is the major approach for the treatment of CO intoxication and they cite references 12 and 13. Note that these references focus on hyperbaric oxygen (typically 2-3atm) and not the more widely accessible (and used) normobaric (1atm) 100% O₂. I assume that the authors want to include both normobaric and hyperbaric 100%O₂ treatment as they also focus part of their study on normobaric 100% oxygen.

The Reviewer is right. We added two additional references about normobaric O₂ ventilation and changed the sentence as follows (page 3 in the revised manuscript): *‘Normobaric (12,13) and hyperbaric O₂ ventilations (14,15) are thus current major approaches for treating CO intoxication.’*

3. Page 8- although the authors conclude that hemoCD1 is safe, there is no toxicology data provided (e.g. markers of heart/liver/kidney damage). The authors refer to previous work (Kitagishi et al (2010) A diatomic molecule receptor that removes CO in a living organism. Angew. Chem. Int. Ed. 49, 1312–1315) but the dosage used in that and other previous work was much lower than in the present work and no toxicology report was provided. With the stability data provided I don’t foresee major problems, but some kind of in vivo toxicology study would be helpful.

We understand the point of the Reviewer. During the animal experiments we carefully observed the organs of the hemoCD1-injected rats and did not find any abnormalities in the organ tissues. The administration of hemoCD1 to animals has been reported by us in several articles (refs 34, 37, 38, 40, 45), where no evident toxicity was found and most of the injected hemoCD1 was excreted in the urine. The dosage used in this study (1.4 and 3.0 mM) was not so different from that of our previous studies (0.6–3.5 mM). We agree with the Reviewer that we need to investigate the toxic effect of hemoCD1 *per se* using markers of tissue and organ damage. However, such experiments will take many resources and a long time and we believe they will be appropriate for a future study. Additionally, the present study is mainly focused on the highly sensitive detection for CO using hemoCD1; therefore, we thought it would be better to focus on the practical use of hemoCD1 as CO antidote using CO-poisoned animal models. We plan to do in the next project that we will focus more on the practical use of hemoCD1 as CO antidote using the CO-poisoned animal models in our next project. To take into account the comment of the Reviewer and to avoid overstatement in the text, we replaced the term “safe” to “effective” in the revised manuscript (end of page 8). Furthermore, we added the following sentence in the Discussion part (page 26 in the revised manuscript): *‘To investigate the practicality of using hemoCD1 as an injectable CO antidote, we plan to expose animals to increasing CO levels and test for organs and tissue damage. In addition, toxicological studies evaluating hemoCD1 per se will be performed in the near future.’*

4. In page 10; the authors should note in the main text that the determination requires dithionite to eliminate any interference from the oxidized hemoCD1 species (this is included in other sections but not here).

We added the role of dithionite for determination of CO in tissues using hemoCD1 in the Result section as follows (page 12 in the revised manuscript): *‘Na₂S₂O₄ added in excess caused denaturation and precipitation of biocomponents that could be readily removed by centrifugation and filtration.’*

5. Page 15; as the authors demonstrate that Hb removes CO from the cell, it actually would indicate that HbCO levels are relevant to assess the extent of CO poisoning (or at least more relevant than tissue CO levels)

We understand the point of the Reviewer and, as already pointed out in point 1, we agree that CO-Hb is still the best available biomarker to define the degree of CO intoxication. However, as we have tried to argue in the present study, it appears that the amount of CO stored in tissues is not relevant to CO-Hb levels. In other words, from our data we infer that the accumulation of CO in tissues does not correlate with CO-Hb levels and, therefore, measurement of CO-Hb alone is less relevant than what is currently believed in assessing the extent of CO poisoning. To stress the point, significant amount of CO is strongly bound to the tissues such as brain and even when CO-Hb levels return to normal after oxygen therapy we still detect high CO levels in this tissue. Our experiments with hepatocytes confirm that there is an interesting

mechanism whereby Co accumulated in cells can be transferred back to Hb, supporting the potential role of Hb in protecting tissue against CO poisoning.

6. Page 16; when the authors indicate that “Thus, once CO is accumulated in tissues it is not easily eliminated by standard treatments used against CO intoxication.” This seems to be a very specific property of the brain and not so much of other tissues. I would emphasize that point as the sentence in its current form is misleading.

We agree with this comment. We have changed the sentence as follows (page 16 in the revised manuscript): *‘However, a significant amount of CO still remained in the brain (cerebrum and cerebellum) even after ventilation with O₂ (Fig. 6C). Thus, once CO is accumulated in the brain it is not easily eliminated by standard treatments used against CO intoxication.’*

7. I am confident that the method proposed by the authors is probably more sensitive than previous accounts; however, there are significant differences in the trends observed by others (Cronje et al (2004). Carbon monoxide actuates O₂-limited heme degradation in the rat brain. Free Radical Biology and Medicine, 37, 1802-1812) and those discrepancies should be discussed in more detail. Note those studies did observe CO accumulation in other organs such as heart but not in brain.

We agree this is an important point to be discussed. We have added the following sentences in the discussion (page 24 in the revised manuscript): *‘A previous investigation on CO biodistribution measured by GC (21) showed that CO was not accumulated in the brain and muscle tissues when animals were exposed to air containing 2500 ppm CO for 45 min. Although this different trend might be attributed in part to the quantification methods (GC vs hemoCD1 assay), we suggest that the difference in the way animals were exposed to CO is a major factor. In fact, in our experiments 400 ppm CO gas was directly introduced into rats via intubation and under these conditions we measured CO accumulation in brain and muscle within 5 min of exposure. This is an interesting observation showing that the distribution of CO in the tissues differs depending on how CO is inhaled.’*

8. In figure 5B and others; the levels in the blood in rest conditions seem way too high. At least in human subjects it is hard to see HbCO levels of 2% or higher in non-smokers. It would be important to validate the HbCO levels with samples quantified by UV-visible spectroscopy

In the original version, we showed CO-Hb% values measured in both arterial and venous blood collected from left and right ventricles (LV and RV). We chose LV and RV because it was suitable for continuous blood sampling without requiring complicated surgery. In general, CO-Hb level in arterial blood was ca. 2% higher than that measured in the venous sample, as other groups have already reported (e.g. *Ann. Emerg. Med.* 25, 481–483 (1995)). In addition, we note that CO-Hb levels in blood from ventricles were somewhat higher than in other veins. We confirmed that the CO-Hb level was ca. 1 % in

other veins such as inferior vena cava (IVC) using the same analyzer. The analysis of blood CO-Hb levels at each site is currently under investigation by one of our co-authors (ATK).

To take in consideration the comment of the Reviewer, only the % of CO-Hb in the venous blood is now reported in the revised manuscript (Figs 5–7). We confirm that the values obtained are quite reproducible, and we highlighted in the manuscript (page 15 in the text and in Supplementary Information for details) the place where the blood was collected from and how the values were determined. We understand the point of the Reviewer and are aware that different basal CO-Hb levels can be obtained depending on the specimen, place of blood collection, methods and instruments used as discussed in *Ann. Clin. Biochem.* **39**, 378–391 (2002), which is cited as ref 10. We hope that the Reviewer agrees with us that what is important is not the basal values as such but the increment (i.e. fold changes) of CO-Hb levels that we detect after treatment with CO gas.

9. In page 18 is noted that “oxy-hemoCD1 infusion during air ventilation (I) fastens the return of CO-Hb to basal levels, while during O₂ ventilation (II) the effect of hemoCD1 infusion is less pronounced.” I would note instead that the effects of O₂ and hemoCD1 seem to be additive, which is to be expected due to their mechanisms of action (but good to be experimentally corroborated)

We agree with this point. We have simplified the sentence as follows (page 19 in the revised manuscript): *The data in Fig. 7A show that oxy-hemoCD1 infusion fastens the return of CO-Hb to basal levels.*

10. Also in page 18 “this... is consistent with the scenario that a high capacity for CO storage of circulating Hb impedes accumulation of excess CO in tissues”. It also indicates that the proteins responsible for CO accumulation in tissues – supposedly myoglobin and Cyt c oxidase- have lower CO affinity than Hb. It begs the question of what is the particular issue with the brain. Is this due to slower diffusion of CO across the brain/blood barrier or is this due to specific proteins in the brain with higher CO affinity, e.g. neuroglobin? Other authors have speculated on heme oxygenase activity (Cronje et al (2004). Carbon monoxide actuates O₂-limited heme degradation in the rat brain. *Free Radical Biology and Medicine*, 37, 1802-1812) but that seems too slow to account for the observed differences in this short timeframe

This is an important point to be discussed. In this regard, we have added a new sentence in the discussion part as follows (page 25 in the revised manuscript): *This different behavior of the brain compared to other tissues might be due to a slower diffusion of CO across the brain/blood barrier and/or due to the existence of specific proteins with high CO affinity such as Ngb.*

11. For the model with the 5 min exposure (Figure 7A) it is not clear why the authors change 2 variables in the air vs oxygen tests, as the hemoCD1 infusion time is changes from 30 min to 15 min. I commend the authors for trying the 100% O₂ model, but note

that the clearance in rats with room air is already much faster than for humans breathing 100% O₂; so other than the comparison of models in figure 6 it would be easier to monitor the changes in CO clearance comparing the effects in the room air model

In protocol I, II, and III, the same amount of hemoCD1 was injected during air/O₂ ventilations regardless of the infusion time. The amounts of CO both in blood and tissue were effectively reduced in all the protocols. At first, we infused the hemoCD1 solution for 30 min (protocol I) then changed to 15 min (II and III) to save the time to collect the organ samples in the case of protocol III. As the Reviewer pointed, a full series of experiments has not been carried out. However, we can conclude that the intravenous injection of hemoCD1 to CO-exposed rats was effective to reduce CO in blood and organ tissues in all protocols.

Concerning the 100% O₂ models shown in Figure 6, it has been proven that both air and O₂ ventilations are insufficient to remove CO from the brain tissues, which motivated us to inject hemoCD1 to rats as demonstrated in Figure 7.

12. In page 21 – “The CO binding affinity tends to be higher in CO-sensing proteins such as neuroglobin (Ngb), NPAS2, CooA, and RcoM”. The function of Ngb is unclear and it is also unknown if Ngb is a CO sensing protein at all; also the CO affinity of NPAS2 is not particularly high. From these 4 proteins only CooA and RcoM are bona fide CO sensors with high CO affinity

This is right. We have changed the sentence to avoid the misleading idea that all these are “CO-sensing proteins” (page 22 in the revised manuscript): “*The CO binding affinity tends to be higher in Hb-R, neuroglobin (Ngb), CooA, and RcoM.*”

13. Page 23. When the authors indicate “We postulate a scenario according to which CO reaching the tissues would gradually transfer to Hb in RBC” this could make sense for the lung, but what would be the route taking CO to the tissues if it is not HbCO as well? It is hard to believe that there is going to be much CO dissolved in plasma as long as there is available Hb around. It would be interesting to estimate such values mathematically

As reported in the literature (ref 11), most of inhaled CO (95%) is thought to be readily captured by circulating Hb and the rest diffuses to organ tissues (page 24 in the revised manuscript). As the Reviewer said, it would be valuable to simulate mathematically the equilibrium among CO bound to Hb, CO dissolved in plasma, and that accumulated in tissues. Small gaseous molecules like CO must diffuse much faster in plasma than to Hb encapsulated in relatively large red blood cells. We think that the diffusion process of CO from lungs to tissues through plasma is not only thermodynamically but also kinetically controlled.

14. Figure 8 is not particularly clear and the left/right duplication in each panel does not help

We agree with this point. Figure 8 has been modified accordingly, along with the points raised by Reviewer #1.

15. In page 23 – “that it is the CO diffused to the tissues and not the fraction bound to Hb that is the principal cause of CO poisoning” However the steady-state amount diffused seems to plateau. If a longer exposure does not increase the tissue CO levels, it would seem that the relevant variable is the time that the tissue is subject to the high CO levels. As the authors point out a role of mitochondrial dysfunction on tissue damage – which seems reasonable- perhaps some markers of mitochondrial dysfunction such as GDF15 may provide a better indication of the CO induced damage

The Reviewer is right. As we demonstrated, a long-time exposure to 400 ppm CO (~80 min) did not further increase tissue CO levels compared to a shorter exposure and did not cause lethality. More concentrated CO gas is required to simulate a CO-poisoning model but at present we are not allowed to use in our institution supplies of more concentrated CO gas. After publishing the current data, we will apply to our institute for permission to use more concentrated CO gas in animal experiments for our next project. We plan to perform a more clinical study using hemoCD1, also measuring the GDF15 marker that the Reviewer suggested. We highly appreciate this kind suggestion.

16. page 24- I agree with the authors regarding the DNS and it may be helpful to comment on some of the trials using oxygen therapy for longer times, even over several days. Although the results are very heterogeneous, the present data would support extended use of oxygen therapy beyond the normalization of HbCO levels

We agree with this point, although from the literature there is not a consensus yet on onset of treatment, doses, time and frequency of exposure to O₂ against CO poisoning. Nevertheless, we have added the following sentence in the Discussion (page 25 in the revised manuscript): *Our data on CO quantification indicates that CO, once stored in the brain, is more difficult to eliminate than from other tissues. This observation supports the idea that extended O₂ ventilation, beyond the normalization of CO-Hb levels, may be necessary to completely remove CO accumulated in brain after intoxication.*

Reply to Reviewer #3:

We thank the Reviewer for her/his kind evaluation on our manuscript. Our reply to all the points raised is as follows.

1. The stability of CO-hemoCD1 is extremely high and thus it can be used as a scavenger of CO. Is it possible to release the CO in the physiological conditions? If not, why does hemoCD1 acts as effective adjuvant to O₂ ventilation to eliminate CO?

It is hard to release CO from CO-hemoCD1 once it is bound to hemoCD1, as demonstrated in the results reported in Figures 2A and 2C. Due to its very high CO binding affinity, hemoCD1 removes CO in vivo and is excreted in the urine as a CO-hemoCD1 complex (ref 38). O₂ and hemoCD1 will remove CO from the tissues via different mechanisms, this is why when used together they are more effective.

2. Closely related to the above question, is it possible to selectively remove CO-hemoCD1 (or CO bound to hemoCD1) after injection to rat?

Injected hemoCD1 to rats will ultimately be eliminated from the body via the kidney and found in urine. These phenomena have been published previously by our group (refs 37 and 38).

3. Although the results are clear, the importance of circulating Hb as a CO scavenger would be somehow a matter of course. If I would understand results correctly, one of the most important findings would be that it is extremely difficult to remove CO once accumulated in cell even with Hb. Therefore, the title, "Highly sensitive quantification of carbon monoxide (CO) in vivo reveals a protective role of circulating hemoglobin in CO intoxication" would be not suitable.

Our study demonstrates that a part of CO accumulated in tissues is hard to remove by Hb circulating in blood. However, it is also clear that accumulation of CO in tissues is mitigated by Hb, as proven by the experiments in Figure 5 showing that more CO was accumulated in tissues in the absence of Hb (ex vivo). Therefore, we think the title reflects the major message of our study.

REVIEWERS' COMMENTS:

Reviewer #1 (remarks to the Author):

The authors addressed all the points and questions I had very carefully including additional experiments and they did the adequate revisions in the manuscript. Hence, this is a very carefully done and scientifically discussed manuscript, which I find very interesting, and so I strongly recommend the revised manuscript for publication.

Reviewer #2 (Remarks to the Author):

The authors have addressed all my comments, I have no further concerns.